# Guardians of Water and Gas Exchange: Adaptive Dynamics of Stomatal Development and Patterning

**DOI:** 10.3390/plants14152405

**Published:** 2025-08-03

**Authors:** Eleni Giannoutsou, Ioannis-Dimosthenis S. Adamakis, Despina Samakovli

**Affiliations:** Section of Botany, Biology Department, National and Kapodistrian University of Athens, 15701 Athens, Greece; egianno@biol.uoa.gr (E.G.); iadamaki@biol.uoa.gr (I.-D.S.A.)

**Keywords:** stomatal development, stomatal differentiation, stomatal lineage, stomatal patterning, hormonal control, abiotic stress

## Abstract

Stomata, highly specialized structures that evolved on the aerial surfaces of plants, play a crucial role in regulating hydration, mitigating the effects of abiotic stress. Stomatal lineage development involves a series of coordinated events, such as initiation, stem cell proliferation, and cell fate determination, ultimately leading to the differentiation of guard cells. While core transcriptional regulators and signaling pathways controlling stomatal cell division and fate determination have been characterized over the past twenty years, the molecular mechanisms linking stomatal development to dynamic environmental cues remain poorly understood. Therefore, stomatal development is considered an active and compelling frontier in plant biology research. On the one hand, this review aims to provide an understanding of the molecular networks governing stomatal ontogenesis, which relies on the activation and function of the transcription factors SPEECHLESS (SPCH), MUTE, and FAMA; the EPF–TMM and ERECTA receptor systems; and downstream MAPK signaling. On the other hand, it synthesizes current discoveries of how hormonal signaling pathways regulate stomatal development in response to environmental changes. As the climate crisis intensifies, the understanding of the complex interplay between stress stimuli and key factors regulating stomatal development may reveal key mechanisms that enhance plant resilience under adverse environmental conditions.

## 1. Introduction

Variations in climate conditions are closely related to the concentration of atmospheric gases, water availability, and atmospheric temperature levels. To adapt to unfavorable environmental and nutritional conditions, plants modulate their developmental programs, often forming new organs as a survival strategy [1]. Leaves, essential for plant development and growth, serve as platforms for photosynthesis and gas exchange. Consequently, plants must optimize their resource allocation, constructing leaves with an optimal structure that balances photosynthetic efficiency with water conservation [1].

Stomata, small pores localized in the epidermis of the aboveground plant organs, mediate the exchange of gases and water with the atmosphere, playing a fundamental role in plant adaptation to environmental changes. The evolution of stomata represents a pivotal innovation in plant colonization of terrestrial environments. Paleobotanical evidence places the earliest confirmed stomata in the late Silurian to early Devonian, approximately 420 million years ago, in the sporophytes of rhyniophytes and trimerophytes—stem groups to all vascular plants [2]. These early stomata consisted of simple guard cell pairs, morphologically closer to the passive pores of bryophyte sporophytes than the dynamic, regulated stomata of modern vascular plants [3]. Over subsequent geological periods, stomatal structure and regulatory complexity evolved in concert with increasing leaf sophistication, from microphylls in lycophytes to megaphylls in ferns, and eventually to the diverse needle and broad-leaf forms of gymnosperms and angiosperms [4]. The anatomical progression of stomatal evolution is paralleled by an equally impressive molecular and regulatory refinement across land plants.

Stomatal characteristics exhibit the strongest influence on gross primary productivity variation, among other leaf anatomical traits. This is based on the fact that, even though stomata occupy just 1–3% of the leaf surface, they mediate over 95% of terrestrial CO_2_ uptake and nearly all water loss in plants. By regulating gas exchange, stomata act as the primary gatekeepers of photosynthetic efficiency and transpiration rates, making them critical determinants of water-use efficiency (WUE) in crops. In terrestrial plants, stomata play a pivotal role in transpiration, directly regulating hydraulic conductance through vascular tissues. This functional interdependence strongly supports the co-evolution of stomata with vascular systems [5]. Therefore, in land plants, water loss is controlled by the number and the size of stomata on the epidermis of aboveground organs. Plants regulate stomatal function—the dynamic opening and closing of stomatal pores—to minimize water loss, positioning guard cells as key operational units in their adaptive response to environmental changes. The precise control of dynamic stomatal closure is based on independent regulatory components, such as the regulation of guard cell turgor and abscisic acid (ABA) responsiveness. These parameters are related to the diversification of terrestrial vegetation that occurred about 480 million years ago, resulting in adaptation to local environmental variations, which radically changed the physiology of many plant species [6]. Previous studies have shown that water balance in guard cells took place about 360 million years ago [7], while the abscisic (ABA)-dependent stomatal regulation mechanism of stomatal pore closure seems to have emerged 400 million years ago [8]. The discovery of genetic variation in stomatal development and function during periods of climate succession can assist in comprehending the mechanisms governing the kinetics of stomatal opening and closure as an adaptation mechanism to unfavorable environments.

This review aims to synthesize current understanding of stomatal adaptations by (i) linking stomatal form and function through analyses of how atmospheric CO_2_ levels, water availability, and temperature regimes influence stomatal density, size, and responsiveness across plant lineages; (ii) evaluating the molecular evolution of key regulatory pathways that control stomatal dynamics; and (iii) identifying genetic variation in stomatal dynamics under rapid climate change to highlight knowledge gaps and propose future research directions for assessing the capacity of stomatal traits to buffer vegetation against environmental extremes.

## 2. Stomatal Development—An Evolutionary Approach

By integrating fossil, anatomical, physiological, and genomic data, we gain a holistic perspective of how a simple pair of protodermal cells evolved into a central regulatory interface for gas exchange, water conservation, and environmental sensing. The stomatal complex is considered a key innovation in plant terrestrialization, while its continued evolution, culminating in the specialized systems of modern angiosperms, underscores its adaptive significance. This supports a mosaic model of stomatal evolution, where both passive and active mechanisms coexisted, with active control becoming more rapid and robust through successive gene recruitment. The evolution of stomata involves both structural refinement and advanced signaling networks, resulting in the dynamic stomatal regulation observed in modern angiosperms, a critical factor in their ecological success.

In the earliest diverging land plants, such as bryophytes (mosses, liverworts, and hornworts), stomata exhibit a surprisingly patchy distribution (Figure 1). Liverworts (Marchantiophyta) appear to lack stomata entirely, with no evidence of their presence even in their fossil relatives. This absence implies either that stomata were absent in the last common ancestor of land plants or that liverworts underwent a secondary loss [9]. In liverworts, air pores are specialized structures that facilitate gas exchange, playing the role of stomata. These air pores are static and open, allowing the continuous diffusion of gases such as CO_2_ and oxygen [10]. Structurally, they are often complex and multicellular, commonly forming barrel-shaped openings surrounded by concentric rings of cells [11,12]. Their development is genetically regulated, reflecting an early adaptation of land plants to terrestrial life [13].

In contrast, both mosses (Bryophyta) and hornworts (Anthocerotophyta) develop stomata (Figure 1), but only on the sporophyte capsule, never on the gametophyte thallus [14]. These sporophytic stomata are often simple, comprising two kidney-shaped guard cells that do not respond dynamically to environmental signals, facilitate passive desiccation, and control spore release [9]. For example, in the model moss *Physcomitrium patens*, stomata are formed during sporophyte maturation; however, guard cells lack a well-developed subsidiary cell complex, remaining open once differentiated. Consequently, their primary function is thought to be the desiccation of sporophyte tissues, enabling capsule opening and spore dispersal, and not gas exchange [15]. Occasionally, stomatal development in hornworts may show rudimentary subsidiary cells [16]; nevertheless, dynamic aperture control has never been documented in any bryophyte [17]. Phylogenetic analysis indicates that, although the last common ancestor of land plants likely possessed a core set of stomatal development and guard-cell-function genes, bryophytes have lost many key components during evolution. This suggests that the stomata found in some modern bryophytes represent rather simplified, ancestral structures that evolved with limited regulatory sophistication. These bryophyte stomata likely arise from pre-patterned epidermal cells with minimal intercellular signaling or environmental sensitivity, which aligns with their restricted role in gas exchange and response to environmental stimuli [18]. Consequently, modern bryophyte stomata appear to represent remnants of a more complex ancestral system that underwent functional reduction in lineages adapting to stable, humid environments.

The lycophytes, encompassing club mosses, spike mosses (*Selaginella*), and quillworts (*Isoetes*), mark a critical stage in land plant evolution as the first vascular plants possessing well-developed stomata on true leaves (microphylls) and stems (Figure 1), which resemble those of seed plants. As such, they are typically flanked by paired guard cells forming a functional pore, further supported by subsidiary cells in some species. Structural differences among lycophyte families reflect adaptive variations, such as sunken stomata in depressions of the epidermis in *Lycopodium*, possibly arising as an evolutionary adaptation to reduce water loss through transpiration [19]. In contrast, *Selaginella* exhibits flush stomata leveled with the epidermis, actively responding to environmental stimuli, indicating the presence of a functional regulatory mechanism [20].

Ferns, and more broadly monilophytes, display the greatest diversity in stomatal architecture among non-seed vascular plants. In true ferns (Polypodiopsida), stomata are primarily anomocytic, characterized by guard cells that lack specialized subsidiary cells (Figure 1). These stomata are typically numerous and evenly distributed across their frequently dissected fronds. On the other hand, Marattiaceae and Ophioglossaceae exhibit paracytic or cyclocytic stomata, featuring distinct subsidiary cell arrangements that differ between lineages. The above morphological variations suggest multiple independent innovations in stomatal architecture long before the emergence of seed plants [21].

In gymnosperms, stomatal development and regulatory mechanisms fully mirror those of angiosperms (Figure 1). Guard cells are kidney-shaped, with variations occurring across lineages. Needles, pinnules, and fan leaves are equipped with clearly defined guard cells forming functional stomatal complexes, often possessing specialized subsidiary cells. Structurally, gymnosperm stomata exhibit a high degree of anatomical complexity and specialization, reflecting adaptations to diverse and often arid terrestrial environments, typically located on the abaxial (lower) surface of leaves or sunken into epidermal crypts. In coniferous species like Pinus and Picea, the deeply sunken stomatal architecture significantly reduces water loss through the combined effects of boundary layer thickening and localized humidity retention [22]. In some species, these pits are additionally protected by wax-secreting trichomes or papillae, further minimizing evaporative demand [23].

These structural adaptations, accompanied by biochemical modifications and reinforced cell walls, highlight the evolutionary challenges gymnosperms overcame to colonize arid, exposed environments. Their stomatal architecture represents an integrated system optimizing water conservation, UV protection, and CO_2_ uptake, resulting in leaves exceptionally adapted to persistent drought and low-nutrient conditions [24]. This sophisticated configuration, inherited and enhanced from early seed plants, established the evolutionary foundation for the more diverse stomatal systems seen in angiosperms.

Angiosperms represent the summit of stomatal evolution, with monocotyledons and particularly grasses exhibiting some of the most sophisticated stomatal systems in plants (Figure 1). Monocot stomata frequently display linear arrays of dumbbell-shaped guard cells flanked by specialized subsidiary cells, enabling exceptionally rapid and precise pore regulation in fluctuating environments. This architectural innovation is supported by an expanded genetic repertoire featuring duplicated genes, distinct expression profiles, and enhanced ABA and light signaling cascades. Together, these adaptations provide unparalleled stomatal control, pointing to the remarkable ecological success and global proliferation of grasses across diverse and often extreme habitats.

## 3. The Development of the Stomatal Cell Lineage

Notably, much of our current understanding of the molecular and genetic mechanisms governing stomatal development comes from detailed analyses in *Arabidopsis thaliana*, where loss- and gain-of-function mutants and transgenic reporters have illuminated key signaling components. In this model eudicot, stomatal development proceeds through a tightly regulated sequence of asymmetric cell divisions and fate transitions, orchestrated by intercellular signaling pathways and cell-to-cell communication. A multifaceted complex of molecular signaling pathways leads to the establishment of a signaling cascade that amplifies and transforms extracellular stimuli into endogenous signals that regulate stomatal shape and distribution in the plant epidermis. During the early stages of stomatal ontogenesis (Figure 2a), a series of cell fate transitions take place; in this context, a protodermal cell, after receiving the appropriate stimuli, turns into a meristemoid mother cell (MMC), which divides asymmetrically to form a smaller meristemoid cell (MC) and a larger stomatal lineage ground cell (SLGC). After a few “amplifying” asymmetric cell divisions, MCs differentiate into guard mother cells (GMCs), which finally divide symmetrically to form guard cell (GC) pairs that surround the stomatal pore with their ventral walls [25,26,27]. Stomatal distribution in the epidermis follows the “one-cell spacing” rule, according to which GC complexes are spaced by pavement cells that facilitate stomatal pores’ active opening and closure [28].

Despite increased efforts being made in recent years, our understanding of the regulatory mechanisms of stomatal development and patterning in cereal grasses remains rudimentary. Monocot stomata (Figure 2b) contain two dumbbell-shaped GCs flanked by two subsidiary cells (SCs), which are developed in parallel rows within defined epidermal cell files from the base towards the tip of the leaf [29,30,31]. Stomatal development in grasses follows a well-defined sequence of stages: (a) establishment of the stomatal row, (b) generation of the guard cell mother cell (GMC) via an initial asymmetric division, (c) establishment of polarity in the subsidiary mother cell (SMC), (d) generation of two subsidiary cells (SCs) via two asymmetric divisions of the adjacent SMCs, (e) formation of two immature guard cells through a symmetric division, and finally (f) maturation of two dumbbell-shaped GCs [29,32,33,34,35,36]. During stomatal ontogenesis in grasses, young GMCs first appear near the base as rectangular cells before transitioning into a square morphology. At this stage, they induce neighboring epidermal cells to differentiate into SMCs. A microtubule (MT)-dependent mechanism subsequently reduces GMC width, triggering localized protrusion of adjacent SMCs toward the developing GMCs [33,37,38,39,40,41]. Prior to asymmetric division, a cascade of polarization events ensures proper SC formation [30].

## 4. The Core Signaling Cascade Governing Stomatal Development

### 4.1. Transcription Factors

In dicots, three basic helix–loop–helix (bHLH) transcription factors, SPEECHLESS (SPCH), MUTE, and FAMA, control a series of cell fate transitions and differentiation events from a protodermal into a guard cell (GC) [26,42,43,44]. Each of these genes plays distinct yet complementary roles driving differentiation events within the stomatal cell lineage (Figure 2c). Their coordinated action at the plant surface ultimately produces mature stomata capable of highly efficient gas and water exchange [45].

More specifically, SPCH, together with INDUCER OF CBF EXPRESSION1 (ICE1), also known as SCREAM (SCRM) [46,47,48], determines cell fate in the stomatal cell lineage (SCL), driving MMC asymmetric cell divisions (Figure 2c) [43]. MUTE, along with ICE1 and SCRM transcription factors, terminates asymmetric cell divisions of meristemoids and enhances their differentiation into GMCs [46,47,49]. MUTE regulates the transition from asymmetric to symmetric divisions in the stomatal lineage by controlling cell cycle progression (Figure 2c). Unlike FAMA, which accumulates in late G2, MUTE peaks during the early G2 phase. This temporal regulation enables MUTE to directly activate cell cycle regulators. Consequently, MUTE loss-of-function results in G1-arrested meristemoids that persistently undergo asymmetric divisions [50].

The successor transcription factor, FAMA, together with ICE1 and SCRM, regulates the final step of stomatal differentiation (Figure 2c) by restricting GMC division and promoting GC differentiation [42]. Interestingly, RNA polymerase II-associated proteins regulate cell fate transitions in the stomatal cell lineage through direct physical interactions with SPCH, MUTE, FAMA, and SCREAM transcription factors [51]. Abnormal regulation of each of the master transcription factors, SPCH, MUTE, and FAMA, has a distinct effect on cell fate specification and the development of the stomatal cell lineage. SPCH overexpression leads to stomatal precursor overproliferation, and overexpression of MUTE induces the formation of GMC-like fates before dividing into paired guard cells [26,43], while gain-of-function *scrm-D* mutants lead to an epidermis full of guard cells [44]. On the other hand, overexpression of FAMA results in trans-differentiation of cells into guard cells without cell division, thereby producing single guard cell stomata that are not functional [42]. In addition, FAMA interacts with the FOUR LIPS (FLP)/MYB88 transcription factor to regulate the symmetric division of GMCs and acts as a negative regulator of CDKB1;1 and CYCA2;3 expression through its direct binding to their promoters, restricting the GMCs to a single division [52,53,54]; this is why loss-of-function FAMA mutants display cell clusters without GC identity [55,56]. Furthermore, loss-of-function FLP and MYB88 induce the formation of clusters of four or more guard cells [57]. Both FAMA and FLP/MYB88 associate with RBR, a negative regulator of cell cycle gene expression and cell proliferation, forming a complex that binds to the promoter of CDKB1;1 [56,58]. Moreover, F-BOX STRESS-INDUCED 4 (FBS4) expressed in epidermal precursor cells associates both with Skp1-Like1 (ASK1), the core subunit of the SCF E3 ubiquitin ligase complex, and with CYCA2;3. Therefore, FBS4 regulates stomatal symmetric divisions at the post-translational level, driving CYCA2;3 degradation via the ubiquitin-26S proteasome pathway [59].

New research shows how stomatal development in the gynoecium is timed with fertilization to enhance floral photosynthesis. Before fertilization, AGAMOUS-SEPALLATA complexes regulate both carpel development and MUTE repression via its first intron, ensuring proper stomatal maturation timing. In parallel, heterodimers of SPCH and SCREAM are already bound to the promoter of *MUTE*. During the progression of gynoecial valve development, the AG and SEP3 protein levels decline, reaching a critical threshold, and can no longer repress the transcription of MUTE. Thereby, *MUTE* expression levels increase, enabling the expression of *FAMA* to form mature stomata. This coincides with fertilization but does not depend on it [60].

Notably, Arabidopsis orthologs SPCH, MUTE, FAMA, ICE, and SCRM2 are found in various grass species. Whereas the FAMA function seems to be conserved, the roles of SPCH and MUTE paralogs have not yet been fully clarified in grasses. Genetic analyses showed that rice and Brachypodium possess two *SPCH* genes, while maize contains three, displaying redundant roles in the initiation of the stomatal lineage [61,62,63,64]. Interestingly, expression of *OsSPCH2* under the control of the *AtSPCH* promoter does not complement the Arabidopsis *spch* mutant phenotype, indicating a degree of functional divergence between *OsSPCH* and *AtSPCH* [65].

In contrast to the AtMUTE ortholog, grass MUTE proteins, characterized by a grass-specific C-terminus, appear to have gained the ability to move from guard mother cells (GMCs) to subsidiary mother cells (SMCs) to promote SMC formation [34]. In addition, grass MUTE proteins contain multiple predicted MAPK phosphorylation sites, which are absent in Arabidopsis MUTE, while they display divergent functions in stomatal development among various grass species, being essential for SC development in maize and rice, but not in Brachypodium [34,36,63]. This differential requirement for MUTE in producing viable stomata opens a new perspective on other elements of the stomatal lineage network to compensate for the specific absence of MUTE.

Recent studies revealed that BdFAMA is necessary and sufficient for specifying stomatal GC fate, while its expression can compensate for the absence of MUTE [66]. On the other hand, while OsFAMA is necessary for the production of functional GCs, it is unable to inhibit cell cycle progression, as osfama mutants display incompletely differentiated GCs and SCs [63,65]. In Brachypodium and rice, grass SCRM (ICE1) orthologs play a crucial role in regulating stomatal development and GMC establishment, asymmetric division of the SMC, and stomata maturation [62]. Furthermore, a conserved physical interaction between OsICE1/SCRM2 and OsSPCH, OsMUTE, and OsFAMA was also noticed [63].

In bryophytes, including mosses, liverworts, and hornworts, a FAMA-like basic helix–loop–helix (bHLH) transcription factor has been identified, while SPCH and MUTE are absent. Interestingly, genomic and transcriptomic analyses in *Selaginella moellendorffii,* a model lycophyte, revealed that MUTE, FAMA, and ICE1/SCRM transcription factors are present, suggesting that significant portions of the core transcriptional regulatory network required for guard cell specification were already in place in the lycophyte lineage. However, SPCH is absent or has diverged beyond recognition, implying either an independent evolutionary path or a reduced requirement for this specific control in lycophytes [67].

### 4.2. Surface Signaling

Stomatal spacing and patterning are controlled by intercellular signaling via peptide ligands, transmembrane receptors, and MAPK modules [68]. Three plant peptide hormones, EPIDERMAL PATTERNING FACTOR1 (EPF1), EPF2, and EPFL9/STOMAGEN, fine-tune proper stomatal patterning [69,70,71,72]. EPF1 and EPF2 act as negative regulators of stomatal development (Figure 2c) [69,70,71]. STOMAGEN, a positive regulator of stomatal development expressed in immature mesophyll cells but not in the epidermis [72,73,74,75,76], competes with EPF1 and EPF2 for receptor binding (Figure 2c) [72]. Experimental evidence showed that the EPF peptides competitively bind to a receptor complex composed of leucine-rich repeat receptor kinase (LRR-RK) family members (Figure 2c), such as ERECTA (ER), ER-LIKE 1 (ERL1), ERL2, and an LRR receptor-like protein, TOO MANY MOUTHS (TMM) [69,70,74,75,76,77]. Recently, EPF1, EPF2, and EPFL9/STOMAGEN were found to regulate the cellular divisions and cell fate transitions [78], having a central role in the establishment of a balance between stomatal development and patterning (Figure 2c), which allows the plant to respond efficiently to environmental stimuli by adjusting water loss and gas exchange [79,80].

The role of ER receptors in stomatal formation is well-established; however, the way that the activated ER receptors are desensitized and recycled is not yet well understood. Plasma membrane receptor homeostasis, which modulates signaling intensity, is maintained through a balance between receptor biosynthesis, trafficking to the cell surface, and activation-induced internalization followed by degradation. Besides their localization at the plasma membrane, ER receptors have also been found in the trans-Golgi network/early endosome (TGN/EE), the prevacuolar compartment, and the tonoplast [81]. Recent findings show that VAP-RELATED SUPPRESSOR OF TMM (VST) proteins positively regulate ER receptor activity by creating contact points between the plasma membrane and the endoplasmic reticulum. The anterograde trafficking of ERL1 and ERL2 receptors from the endoplasmic reticulum to the plasma membrane requires chaperone-mediated folding by the SDF2-ERdj3B-BiP complex [82]. Following ligand perception, activated ERL1 undergoes endocytosis and is subsequently sorted for vacuolar degradation via the multivesicular body/late endosome (MVB/LE) pathway [82].

A recent study shed light on the recycling of ER receptors, showing that two closely related plant U-box ubiquitin E3 ligases, PUB30 and PUB31, serve as attenuators of ER signaling during stomatal and inflorescence/pedicel development [83]. Upon EPF2 and EPF6 perception by ER, the association of the ER-family/TMM receptor with SOMATIC EMBRYOGENESIS RECEPTOR KINASE 1, SERK2, SERK3 (also known as BAK1), and SERK4 is promoted to convey the cellular signal [72,76] (see below). Upon this activation step, BAK1 phosphorylates PUB30/31, resulting in a stronger association of PUB30/31 with ER and BAK1 [81]. In this molecular mechanism, BAK1 has the role of both co-receptor and scaffold protein, which recruits PUB30/31 to ubiquitinate ER, but not BAK1 itself, leading to ER degradation and signaling attenuation [83]. In addition, Novel Regulators of Plasma Membrane Proteins (NRPM), peripheral membrane proteins expressed in epidermal and mesophyll cells, function as negative regulators of stomatal development. Positioned downstream of EPF1/EPF2 but upstream of the YDA MAPK cascade, NRPMs play a crucial role in maintaining ERECTA-family receptors at the plasma membrane [81]. Notably, NRPM depletion leads to receptor retention within the endomembrane system, impairing proper membrane localization [81].

Interestingly, experimental data show that stomatal patterning and size differ in fully expanded Arabidopsis true leaves, indicating that stomatal density depends on the leaf number [84]. In addition, stomatal density in the abaxial epidermis is higher compared to stomatal density in the adaxial epidermis. Two independent molecular mechanisms control stomatal development in the abaxial and adaxial epidermis. Whereas in the abaxial epidermis SDD1, EPF1/2, and TMM play important roles in the suppression of stomatal development, ERL2 and ABA signaling pathways suppress stomatal ontogenesis in the adaxial epidermis [84].

In grasses, secretory peptides and downstream MAPK participate in regulating stomatal development and patterning. In major cereal crops, overexpression of the grass *EPF1* gene has been shown to arrest stomatal production [78,85], whereas knockout of a rice ortholog of Arabidopsis *OsEPF9a* results in a severe reduction in stomata density [86], suggesting the coexistence of negative and positive signaling peptides regulating stomata formation in grasses. Overexpression of *EPF1* genes (*HvEPF1*, *OsEPF1*, *TaEPF1*) inhibits GMC formation and arrests their development before SMCs form, reducing stomatal density and improving water-use efficiency without affecting grain yield [78,85,87,88]. Gene duplications of *EPF/EPFL9* are found in several grasses and may have contributed to the evolution and functional diversity of EPF signaling [89].

Interestingly, lycophyte genomes contain genes encoding EPF-family peptides and TMM-like receptors, indicating that peptide-mediated stomatal spacing mechanisms were likely already functional in the early tracheophytes and contributed to the emergence of regulated and spaced stomata well before the divergence of euphyllophytes (ferns and seed plants) [67]. The *Ceratopteris* genome also contains homologs of TMM and ERECTA-family receptor-like kinases [90]. Importantly, EPF-like peptides were found to be expressed in neighboring non-stomatal epidermal cells, consistent with their role in enforcing the “one-cell spacing” rule. This suggests that peptide–receptor signaling pathways are functional in ferns, challenging earlier assumptions that such regulatory sophistication is present only in flowering plants, and highlights the early origin of intercellular signaling mechanisms in vascular plant evolution [90,91]. Gymnosperms, including Picea, Pinus, and Gnetum, retain TMM and EPF peptides and ERECTA-family receptor-like kinases in their genomes [92].

### 4.3. Downstream Signaling Events

A series of sequential phosphorylation events through a MAPK signaling cascade act downstream of EPF and the ER/TMM/SERK ligand–receptor complex to regulate stomatal development in Arabidopsis [93]. This cascade starts with YODA (YDA), a MAPKKK, followed by MKK4/5 and MPK3/6 [94] to regulate multiple developmental processes, such as asymmetric division and stomatal patterning (Figure 2c). Loss of function of the MAPK cascade genes results in overproduction of stomata and clustering, while gain-of-function mutants show inhibition of stomatal development [94]. Induced or repressed activity of MPK3/6 in the epidermis or mesophyll cells changes the dynamics of stomatal cell lineage proliferation. STOMAGEN acts downstream of MPK3/6 in mesophyll cells and upstream in epidermal cells during stomatal development, which allows MPK3/6 to modulate stomatal development according to the signals received from the mesophyll cells [95].

One of the main targets of the MAPK signaling cascade is the SPCH protein. MPK3 and MPK6 phosphorylate SPCH, leading to changes in SPCH protein stability or affinity with other partners [49]. SPCH can also be phosphorylated by multiple different classes of protein kinases, including BRASSINOSTEROID INSENSITIVE2 (BIN2) [96] and cyclin-dependent kinase A1 (CDKA;1) [97], pointing out that SPCH activity is the key regulatory point where environmental and physiological cues intersect.

In *Brachypodium*, the loss-of-function *Bdyda1* mutant displays a large group of stomata placed in contact in a single row, breaking the stomatal spacing rule; however, no ectopic stomatal rows were found, suggesting that cell fate determination in the stomatal lineage is not affected [98]. Experiments in *Ceratopteris* have shown that MAPK signaling is essential for correct stomatal development in ferns, as depletion of MAPK activity disrupts normal guard cell spacing and morphogenesis. Moreover, the presence and activity of CDKB-type cyclins indicate that ferns have retained complete and operational division machinery for stomatal formation [90]. Converging evidence from comparative genomics reinforces that gymnosperms also retain a complete MAPK cascade. The above highlights the evolutionary depth and resilience of stomatal regulatory networks in land plants.

### 4.4. Polarity Mechanisms and Stomatal Development

Cell polarity plays a pivotal role in plant development and growth, as stem cell division and differentiation are mediated by proteins polarized to cortical domains at the plasma membrane. Compelling evidence highlights the role of protein polarization during asymmetric cell division, revealing an additional level of temporal regulation within the stomatal cell lineage (Figure 3a). In Arabidopsis, before each asymmetric cell division, BREAKING OF ASYMMETRY IN THE STOMATAL LINEAGE (BASL), which is first localized at the nucleus, moves to the distal site of the newly formed division plane, leading to cell fate specification [99]. After cell division, the polarity of BASL is maintained only in SLGC, driving its exit from the stomatal cell lineage [99].

Furthermore, BASL tightly controls *SPCH* expression during the asymmetric cell divisions by recruiting the MAPK signaling module [99,100]. MPK3/6 phosphorylates BASL and regulates its localization and function [100]. Then BASL interacts with YDA, acting as a scaffold that polarly localizes the MAPK cascade at the cell cortex in SLGCs [100]. POLAR LOCALIZATION DURING ASYMMETRIC DIVISION AND REDISTRIBUTION (POLAR) proteins function as a scaffold to polarly restrict BIN2 along with BASL at the cell cortex and mediate the attenuation of the MAPK cascade (Figure 3b), suggesting that BIN2 acts together with POLAR to regulate its other activity during stomatal development [28,101].

The specification of the polarization domain of BASL relies on interactions with the cytoskeleton. The polarized BASL cortical domain is characterized by local depletion of stable microtubules. BASL polarity influences the orientation of divisions through the control of preprophase band placement and thereby the orientation of cell divisions and subsequently cell fate. Furthermore, BASL associates with PRAF/RLD, which is functionally connected with Arf GEF GNOM-dependent endosomal trafficking [102]. Therefore, positive feedback between BASL and membrane trafficking through PRAF is responsible for the establishment of the intrinsic polarization of BASL at the membrane, mediating asymmetric cell divisions in the stomatal cell lineage [102]. BASL and POLAR mimic the behaviors of the PAR (partitioning defective) protein complexes, which asymmetrically localize to induce “symmetry breaking” of the mother cell and direct differential segregation of the cell fate determinants into two daughter cells in animal systems. Notably, phosphorylation by BIN2 negatively controls the activity of YDA, MKK4/5, and SPCH [96,103,104].

A unique feature of stomatal development in grasses is the formation of SCs that allow the swift transport of ions and solutes to GCs. SMCs are formed on both sides of a GMC through asymmetric divisions. The first indication of SMC polarization is the detection of Brk1, a small subunit of the SCAR/WAVE regulatory complex (SWRC) homologous to mammalian HSPC300 [105,106]. In *Zea mays*, Brick (Brk) proteins play a crucial role in SC formation, as many SMCs in brk mutants exhibit disrupted polarization and misoriented cell division planes, leading to impaired asymmetric SMC divisions [105,106,107]. The SWRC exhibits polarized localization, initially accumulating at the contact region between SMCs and young GMCs. It persists at the polar site of SMCs and is later inherited by young SCs [108]. For SWRC polarization to occur, PANGLOSS2 (PAN2) is recruited to the polar SMC site, where it subsequently promotes the accumulation of PANGLOSS1 (PAN1). The localization of both PAN1 and PAN2 at this site is a critical step in SMC polarization [109]. In *pan* mutants, many SMCs fail to form actin filament (AF) patches, and their nuclei remain unpolarized, ultimately disrupting the orientation of the cell division plane [107,109,110,111]. PAN1 interacts with the small guanosine triphosphatases RHO OF PLANTS (ROP2/9), both of which colocalize at the GMC contact site [110]. Activation of the SWRC complex leads to RIC4 recruitment, a component of the ARP2/3 complex [108], suggesting that ROPs play an active role in regulating F-actin dynamics and the polar positioning of the nucleus [105,108,110]. This process ultimately promotes asymmetric division in SMCs. The formation of AF patches is a fundamental aspect of SMC polarization and is considered a key marker of polarity. While its precise role remains unclear, the AF patch may not directly drive nuclear migration toward the GMC [112]. Instead, it could be involved in polarized vesicle trafficking [110] or in protecting the SMC plasmalemma at the polar site from mechanical stress caused by localized SMC expansion toward the GMC [30].

Recent studies in Brachypodium, rice, and maize have demonstrated that MUTE proteins moving from GMCs to neighboring SMCs mediate the establishment of SMC identity [34,36]. Interestingly, ZmMUTE directly regulates the expression of *PAN1* and *PAN2* in SMCs [34,36], whereas OsMUTE and its downstream transcriptional targets activate the expression of peptide signals in GMCs [64]. Despite these advances, identifying the precise signal from GMCs that establishes SMC polarity remains a critical goal for future research.

## 5. The Role of Hormonal Signaling in Stomatal Cell Lineage Proliferation

Stomatal development and function are vital for plant growth and survival, as stomata regulate photosynthesis while optimizing water-use efficiency (WUE). This is the reason why stomatal development is under the control of complex and coordinated regulatory networks, in which, besides the transcription factors SPCH, MUTE, and FAMA, hormonal signaling and other proteins play essential roles in cell fate specification and differentiation [112,113,114].

### 5.1. Brassinosteroids and Stomatal Formation

Brassinosteroid (BR) signaling controls plant growth by activating the plasma membrane BRASSINOSTEROID INSENSITIVE 1 (BRI1) upon the binding of brassinolide (BL) [115]. This releases the inhibitor BRI1 KINASE INHIBITOR1 (BKI1), recruits its co-receptor BRI1-ASSOCIATED RECEPTOR KINASE1 (BAK1), and triggers trans-phosphorylation to dephosphorylate GSK3/SHAGGY BRASSINOSTEROID INSENSITIVE kinase 2 (BIN2). BIN2 functions as a negative regulator of downstream nuclear BR signaling events. In the presence of BL, BIN2 is inactive and can no longer phosphorylate and inactivate the master transcription factors BRI1-EMS-SUPPRESSOR1 (BES1) and BRASSINAZOLE RESISTANT 1 (BZR1). The active BES1 and BZR1, therefore, can induce the transcription of genes affected by BL signaling that regulate a broad spectrum of plant developmental processes [115]. Several lines of evidence suggest a significant role for the BR signaling pathway in stomatal development (Figure 4): a. Overexpression of BRI1 induces stomatal production, while *bri1* mutants show changes in stomatal distribution, affecting the stomatal index and altered stomata size [96]. b. Recently, EPF1, EPF2, and EPFL9/STOMAGEN were found to regulate cellular divisions and cell fate transitions [78,116]. This binding promotes the association of ER-family/TMM receptor complex with SOMATIC EMBRYOGENESIS RECEPTOR KINASE 1, SERK2, SERK3 (also known as BAK1), and SERK4 to convey the cellular signal [76,117,118,119]. Notably, only the higher-order *serk* mutants exhibit severe stomatal clustering, while SERKs have been shown to associate with ER/ERLs in a ligand-induced mode [118].

BIN2 directly phosphorylates YDA, MKK4, MKK5, and SPCH [120] to fine-tune asymmetric cell divisions in the stomatal cell lineage (Figure 4a). A recent study uncovered a novel role for GSK3-like kinases, particularly BIN2, in stomatal lineage progression. The findings suggest that the precise formation, accumulation, and localization of BIN2, BASL, and the POLAR complex are critical for cell fate specification in the stomatal lineage, with the plant-specific POLAR protein functioning as a scaffold [101]. The study revealed that in meristemoid mother cells (MMCs), BIN2 is barely detectable, whereas POLAR is highly expressed. As MMCs mature, BIN2 levels rise, and BIN2—alongside MAPKs—initiates BASL polarization. This leads to the accumulation of both BIN2 and POLAR at the polarity sites where BASL is localized. Consequently, BIN2 and the MAPK cascade are recruited to the cytosol, alleviating SPCH inhibition in the nucleus. Sustained SPCH activity maintains *POLAR* and *BASL* transcription, thereby promoting asymmetric cell division [101]. In the resulting stomatal lineage ground cell (SLGC), POLAR expression persists, and the BIN2-POLAR-BASL polarity complex accumulates at the opposite pole, enabling a properly spaced asymmetric division. Conversely, reduced POLAR expression in SLGCs redirects BIN2 to the nucleus, where it cooperates with the MAPK cascade to inactivate and degrade SPCH. This shift drives pavement cell formation and differentiation [101]. A recent study elucidated how scaffold proteins confer cellular signal specificity to BIN2, modulating its function to either promote or restrict asymmetric stomatal division. This regulation depends on BIN2’s subcellular localization, which is determined by its interaction with the stomatal-lineage-specific scaffold protein POLAR or its closest homolog, POLAR-LIKE1 (PIL1) [121]. When BIN2 associates with POLAR, it becomes resistant to BR-mediated inactivation, resulting in attenuation of BR signaling in asymmetric division precursors [121]. In contrast, BR signaling remains active in differentiating epidermal cells, where POLAR is absent.

The BSU1 (bri1 Suppressor 1) phosphatase plays a well-established role in the brassinosteroid (BR) signaling pathway by dephosphorylating and inactivating the kinase BIN2 [115]. A recent study further expanded this paradigm by revealing that the BSU1-like (BSL) family of Ser/Thr protein phosphatases also promotes asymmetric stomatal divisions. These phosphatases operate through a kinase-dependent mechanism, ensuring proper cell fate specification in both daughter cells [122]. BSL1 is polarized at the cortex of the MMC through its association with the BASL polarity complex at the onset of mitosis, monitoring the division plane asymmetry. This association with the polarity complex controls the partitioning of BIN2 in the nucleus, activates YDA MAPK signaling, and regulates the cell cycle [122]. Notably, the polarized localization of BSL1 is inherited by the differentiating daughter cell, where it suppresses further divisions and promotes cell fate determination. This mechanism serves as a spatiotemporal molecular switch, ensuring proper cell fate specification in asymmetrically dividing daughter cells [122]. Furthermore, other BSL family members (BSL2, BSL3, and BSU1) suppress MPK6 activity in the nucleus, promoting stomatal formation [123]. Together, these findings underscore the critical role of spatiotemporal regulation of MAPK signaling in stomatal lineage determination, mediated by the distinct functions of BSL phosphatases [123].

Interestingly, gain-of-function mutants of BR master transcription factors such as *bri1-ems-suppressor1* (*bes1-D*) and *brassinazole-resistant1* (*bzr1-D*) do not display any differences in stomatal formation compared to wild-type [46,96]. These findings support the conclusion that BIN2 represses stomatal development independently of its canonical targets BES1 and BZR1 [96]. However, earlier studies reported contrasting transcriptional effects in *bzr1-1D* and *bes1-D* backgrounds: while *MKKK9* expression was upregulated, *FAMA* expression was suppressed [42,49]. These observations were recently confirmed by an independent study [124], prompting a revised model for BR-mediated stomatal regulation. According to this updated framework, (i) BRs inhibit stomatal development through the repression of BIN2. (ii) The repressed BIN2 fails to phosphorylate (and thus inactivate) BZR1/BES1, allowing these transcription factors to accumulate. (iii) BZR1/BES1 upregulate *MKK9*, which blocks the meristemoid-to-GMC transition. (iv) Concurrently, *FAMA* repression impedes GMC-to-GC differentiation, particularly in etiolated cotyledons [124].

Brassinosteroid (BR) signaling exerts opposing effects on stomatal development in cotyledons and hypocotyls (Figure 4a). In contrast to cotyledons, BRs promote stomatal development in hypocotyls via a BIN2-dependent but BES1/BZR1-independent mechanism [96]. This tissue-specific regulation arises because hypocotyl stomatal patterning shares genetic components with root epidermal development [125]. In this view, TMM functions differently in hypocotyls, as *tmm* hypocotyls are devoid of stomata, displaying arrested meristemoids [126], while BL treatment increases the production of meristemoids in the *tmm* hypocotyls [96,127]. Intriguingly, mutations in CHALLAH (CHAL) signal peptides expressed in internal hypocotyl tissues could suppress stomatal overproduction in hypocotyls without any effect on stomatal development in cotyledons [128,129]. These findings led to the following proposed model: TMM enhances EPF1/2 signaling through ER family receptors at the epidermis of all the aerial organs producing stomata; however, it reduces CHAL signaling in the hypocotyl and stem [128]. In *tmm* mutants, EPF1/2 downregulation results in stomatal overproduction in the cotyledons, and CHAL upregulation leads to suppression of stomatal development in hypocotyls and stem [128]. Downregulation of EPF1/2 results in reduced activity of the YDA cascade in hypocotyls under low BR levels, which makes BIN2-mediated regulation of SPCH the main molecular mechanism controlling stomatal initiation [130].

### 5.2. Auxin and Stomatal Development

Auxin acts as a master regulator of plant development, orchestrating critical processes throughout the plant life cycle. It plays central roles in embryogenesis, polar axis establishment, seedling development, and organ formation [131]. Auxin-mediated plant growth is achieved through alterations in the active form of hormones that affect different tissues depending on their sensitivity. Among others, auxin controls cell division and differentiation [131], while it has recently been connected to stomatal development. This notion relies on the following findings: It was shown that genetic or pharmacological inhibition of the PIN-FORMED (PIN) family of auxin efflux carriers, causing problems in polar auxin transport, results in stomatal clustering. In addition, analysis of active auxin levels revealed that auxin activity is higher in young meristemoids and declines before the differentiation of meristemoids into GMCs, which is followed by the accumulation of PIN3 at the plasma membrane of late meristemoids and GMCs [132]. This suggests that auxin controls the switch from asymmetric to symmetric cell division, controlling stomatal differentiation (Figure 5a). The hypothesis that auxin acts as a negative regulator of stomatal differentiation was further supported by experimental work showing that exogenous application of auxin reduces the stomatal index and, therefore, the number of epidermal cells differentiating into guard cells [133]. In addition, disruption of the TAA1/TAR auxin biosynthetic pathway leads to stomatal clustering [134].

The aforementioned experimental data demonstrate that auxin biosynthesis, transport, and nuclear auxin signaling fine-tune stomatal development (Figure 5a). Auxin signaling is perceived in the nucleus by TRANSPORT INHIBITOR RESPONSE 1 (TIR1) and its co-receptor AUXIN-BINDING F-BOX (AFB). Auxin early response is then mediated through AUXIN/INDOLEACETIC ACID (Aux/IAA), short-lived nuclear proteins acting as transcriptional repressors. The auxin-induced degradation of Aux/IAA regulates the transcriptional function of AUXIN RESPONSE FACTORs (ARFs). Stomatal formation is affected by TIR1/AFB, ARF MONOPTEROS (MP), and Aux/IAA BODENLOS (BDL), since genetic inhibition of the above induces stomatal clustering and increased stomatal precursors [132,133,134]. Auxin has been linked mostly to STOMAGEN transcriptional regulation, as its expression is directly controlled by stomatal MP [133]. STOMAGEN repression under high auxin levels suppresses stomatal ontogenesis by activating the TMM/ER-MAPK module. However, the possibility that additional auxin targets besides STOMAGEN participate in stomatal development cannot be excluded. Auxin also regulates stomatal distribution and spacing, most probably through the control of asymmetric cell divisions; nevertheless, the molecular mechanism through which this happens remains elusive. Therefore, understanding the way that auxin functions to determine the polarity of cell divisions in the stomatal lineage would be extremely interesting.

In *Zea mays*, emerging evidence demonstrates that auxin functions as a key regulator of stomatal subsidiary mother cell (SMC) polarization and asymmetric division [135,136]. Time-course analyses reveal dynamic auxin accumulation specifically at GMC-SMC contact sites within the protoderm [136], supporting the hypothesis that GMCs act as localized auxin sources. Exogenous application of auxin near the leaf meristem effectively induces complete stomatal development processes, contrary to pharmacological inhibition of auxin transport that disrupts polarization and division events [136]. In *Zea mays* leaves, PIN1 exhibits striking polar localization patterns, as near the basal leaf meristem, where young guard mother cells (GMCs) are forming, PIN1 shows pronounced accumulation at transverse cell walls across all protodermal cell types, including GMCs [135]. This polarized distribution contrasts sharply with the nearly absent PIN1 signal at lateral cell walls, demonstrating strict directional control of auxin transport during stomatal development. In more differentiated protodermal regions, a notable shift in PIN1 carrier distribution occurs in advanced GMCs: the plasmalemma adjacent to their lateral cell walls, as well as that of the neighboring SMCs, becomes selectively enriched in PIN1 carriers. At the stage when GMCs act as local auxin sources, PIN1 accumulation is more prominent along the lateral GMC cell walls than along the transverse walls [135]. This redistribution of PIN1 in advanced GMCs likely facilitates preferential auxin transport through the apoplast between GMCs and SMCs, resulting in auxin accumulation at the polar site of the SMC [136]. These findings further support the view that auxin acts as an inductive signal emitted by the GMCs and transported toward the SMCs through the asymmetric distribution of auxin carriers.

### 5.3. Cytokinin and Stomatal Development

Cytokinin is a growth regulator controlling diverse processes during plant development, such as plant growth and defense against biotic and abiotic stresses. Cytokinin plays an essential role in the control of cell differentiation and proliferation (Figure 5b), affecting vascular patterning and stomatal formation during leaf development. Biosynthetic and catabolic pathways control the levels of cytokinin in plant cells. The cytokinin signaling pathway is initiated at the plasma membrane by membrane-localized receptors, which, when activated, trigger a series of phosphorylation events to modulate gene expression [137]. Cytokinin-biosynthesis-defective mutants, such as *ipt1357*, as well as cytokinin signaling mutants, such as ARABIDOPSIS HISTIDINE KINASE (AHK) receptor higher-order mutants (*ahk2 ahk3 cre1*), display a significant decrease in asymmetric cell divisions, particularly in spacing and amplifying divisions [138]. Moreover, SPCH functions as an effector of cytokinin signaling, inducing the expression of type-A RESPONSE REGULATOR 16 (ARR16; Figure 5b).

A complex negative regulatory loop is constituted by ARR16 and ARR17, as well as CLAVATA3/ESR-RELATED 9 (CLE9) and CLE10, to fine-tune SLGC spacing divisions [138]. ARR16 and ARR17 are expressed in meristemoids, co-localizing with SPCH. In addition, CLE9 and CLE10 peptides derived from GMCs and developing stomata suppress divisions in the neighboring SLGCs. In summary, in the presence of cytokinins, SPCH induces ARR16/17 and CLE9/10, specifically expressed in the stomatal cell lineage, which antagonizes both cytokinin signaling as well as SPCH-dependent spacing divisions in SLGCs. In this way, cytokinin controls the percentages of stomata and pavement cells on the leaf epidermis [138]. However, the experimental evidence for cytokinin-mediated regulation of stomatal development remains limited. For example, the signaling intermediates that directly control SPCH are not clear yet. The role of type B-ARR TFs, positive regulators of cytokinin signaling, in stomatal development is unexplored.

Interestingly, studies have shown that high cytokinin levels increase leaf size, promoting biomass production [139], while low cytokinin levels that reduce cell divisions in the epidermis result in decreased transpiration, improving drought tolerance in tomato plants [140]. In addition, drought tolerance has been linked to the drought-inducible C-type ARR22 in the stomatal cell lineage [141], while the upregulation of SPCH transcript levels after cytokinin application was abolished in ARR22-overexpressing plants [142]. Taken together, these findings suggest that modulating cytokinin levels or signaling in the stomatal cell lineage could be a promising strategy to conditionally regulate stomatal development, offering potential agronomic benefits.

### 5.4. Ethylene and Stomatal Development

Ethylene is a plant hormone involved in aging and stress responses [143,144]. It controls a broad spectrum of physiological responses through its interaction with auxin [145,146,147,148]. It has been reported that ethylene induces auxin biosynthesis in root and leaf epidermal cells while inhibiting stem cell divisions in the shoot apical meristem [149]. Recently, it was shown that CTR1, a Raf-like kinase that couples with ethylene receptors to inhibit the downstream ethylene signaling cascade [150], increases the stomatal index on the cotyledon epidermis (Figure 5c) [151]. Ethylene signaling reduces the polarity of BREVIS RADIXLIKE2 (BRXL2), a polarity protein involved in amplifying asymmetric cell divisions, thus inhibiting the renewal of stomatal lineage stem cells [151]. In addition, 1-aminocyclopropane-1-carboxylic acid (ACC), the precursor of ethylene, promotes the symmetric division of GMCs by suppressing the activity of FAMA and FLP/MYB88 in an ethylene-independent manner (Figure 5c) [152]. Blocking the biosynthesis of ACC resulted in the formation of single-cell guard cells in both cotyledons and hypocotyls [152]. Interestingly, the application of ACC induced the division of GMCs in *fama* and *flpmyb88* mutants through the upregulation of CDKB1;1 and CYCA2;3; treatments with ethylene synthesis inhibitors did not lead to the same phenotype, supporting the notion that the regulation of GMC symmetric division by ACC is not ethylene-dependent [152].

### 5.5. Gibberellins and Stomatal Development

GA is a growth-promoting hormone regulating many developmental processes [153]. It has been reported that GA controls stomatal production specifically in hypocotyls, as *ga1-1* mutants display no stomata in hypocotyls, contrary to cotyledons, where stomatal formation was unaffected [154]. The essential role of GA in the development of stomata in the hypocotyl epidermis has been connected to its essential role in the regulation of hypocotyl elongation, as well as to the different developmental pathways controlling stomatal formation in hypocotyls. Nevertheless, the molecular mechanism of action of GA in the regulation of hypocotyl stomatal formation has not been clarified yet. Experimental evidence suggests that the GA signaling pathway interacts with the BR signaling cascade and, more specifically, acts upstream of/or directly with BRI1 to control stomatal development in hypocotyls [155].

### 5.6. Jasmonate and Stomatal Development

Jasmonates (JAs), a class of lipid-derived stress hormones, are involved in the regulation of many physiological processes and defense responses against biotic and abiotic stress [156]. During plant development, JAs modulate root elongation and gravitropism, trichome development, anthocyanin biosynthesis, senescence, male fertility and flowering, seed development, primary and secondary metabolism, and germination [157].

Under control conditions, JA levels appear to be low, but in the presence of environmental or developmental stimuli, they increase [157]. The JA signaling pathway initiates with the F-box protein CORONATINE INSENSITIVE1 (COI1), which forms the E3 ubiquitin ligase SCFCOI1 that results in the degradation of JASMONATE ZIM-DOMAIN (JAZ) proteins [158]. JAZ proteins are repressors of JA signaling, as they interact with downstream bHLH-like MYC transcription factors, like MYC2-4, which facilitate the expression of multiple genes involved in JA responses [157,158]. In Arabidopsis, plants deficient in JAs exhibit significant changes in stomatal development [159], while exogenous application of JAs results in repressed formation of stomata in wild-type plants (Figure 4c), as well as in mutants displaying stomatal overproduction [159]. It was shown that MYC transcription factors acting upstream of the master transcription factors SPCH, MUTE, and FAMA repress stomatal formation [160]. Recent reports demonstrate that JAs lead to decreased numbers and sizes of cells by blocking the cell cycle at the G1 phase before entering the S phase [161]. In this context, a plant-specific cyclin-dependent kinase inhibitor, SIAMESE-RELATED4 (SMR4), which is a transcriptional target of MUTE, was recently characterized, providing insights into the molecular network that controls the switch from proliferation to differentiation within the stomatal cell lineage [162]. SMR4 physically and functionally interacts with CYCD3;1 to prolong the G1 phase during asymmetric cell division, while it does not associate with CYCD5;1, permitting symmetric cell divisions [162]. The above findings support the hypothesis that JAs negatively regulate stomatal development through the transcriptional control of genes important for endoreduplication and DNA replication mechanisms that directly regulate cell cycle progression. However, the precise molecular mechanism that connects JAs with the cell cycle and transcription factors controlling stomatal ontogenesis has yet to be revealed. In addition, since JAs are mainly characterized as stress-induced hormones, it is important to shed light on the molecular network in which JA signaling intersects with various environmental factors to control stomatal development in response to exogenous and endogenous stimuli.

### 5.7. ABA and Stomatal Development

Although the role of ABA has been thoroughly analyzed in stomatal opening and closure, little is known about its role in the control of stomatal development. Studies have shown that ABA promotes pavement cell enlargement in Arabidopsis and decreased levels of ABA promote the formation of MCs and MMCs (Figure 4b), suggesting that ABA represses the expression of *SPCH* and *MUTE* transcription factors, thereby blocking stomatal cell lineage initiation and specification [163,164]. Recent studies showed that SnRK2 kinase, a component of ABA signaling, phosphorylates specific residues of the SPCH protein, resulting in the inactivation of SPCH transcriptional activity and, therefore, stomatal production [165]. Furthermore, induced expression of the ABA-responsive transcription factor HOMEODOMAIN GLABROUS11 (HDG11), which results in increased ABA levels, activates ERECTA, indicating a multitier mode of action of ABA in the regulation of stomatal formation [166,167]. In addition, XERICO (XER), a stress-responsive RING E3 ubiquitin ligase that increases ABA levels through the inhibition of ABA catabolism, is involved in stomatal development [168]. Stomatal clustering was observed in mutants deficient in ABA, such as *xer-1*, *xer-2*, and *aba 2-2*, while exogenous application of ABA or inhibition of ABA catabolism alleviated the stomatal clustering phenotype, showing that ABA regulates stomatal distribution and patterning [168]. Genetic analysis demonstrated that XER functions upstream of the EPF2-SPCH pathway in parallel to EPF1 to enable proper stomatal spacing [168].

Besides its crucial role in regulating stomatal formation through protein phosphorylation dynamics, ABA has recently been linked to NAD^+^ metabolism. The biosynthesis of ABA depends on NAD^+^, as zeaxanthin epoxidase (ABA1) converts zeaxanthin into the ABA precursors antheraxanthin and violaxanthin, while xanthoxin dehydrogenase (ABA2)—which further converts xanthoxin into abscisic aldehyde—requires NAD^+^ as a cofactor [169]. Two lines of evidence demonstrate the essential role of NAD^+^ metabolism in guard cell production: NAD^+^ transporter mutants (*ndt1* and *ndt2*) exhibit fewer meristemoid cells (MCs) and guard mother cells (GMCs), resulting in a reduced stomatal index. Moreover, exogenous ABA and NAD^+^ application in these mutants does not alter the expression of the master stomatal regulators SPCH, MUTE, and FAMA [163,170,171]. Thus, NAD^+^ metabolism influences stomatal development by modulating ABA signaling and the expression of key stomatal genes. However, whether NAD^+^ homeostasis also regulates stomatal ontogenesis independently of ABA remains unknown.

## 6. The Impact of Environmental Factors on Stomatal Development

Environmental signals influence diverse physiological processes that govern plant development. As stomata are critical for key functions such as photosynthesis and nutrient uptake, their development is tightly regulated by environmental cues [171] to optimize the balance between photosynthetic efficiency and transpiration under varying conditions [172]. Key factors modulating stomatal ontogenesis—including stomatal density, size, and index—are light intensity, water availability, temperature, and atmospheric CO_2_ concentration [173,174].

### 6.1. Stomatal Development and Light

Light, by acting as a developmental signal and an energy source, is considered indispensable to plant growth and development. Stomatal development is positively regulated by light, as it directly regulates cell differentiation in plants, acting on undifferentiated cells like MCs (Figure 6a) [174,175,176]. Light perception relies on the function of photoreceptors such as phytochromes and cryptochromes [177], which, upon activation, induce the expression of genes involved in cell differentiation, which in turn increases guard cell production [175,178,179]. Therefore, genetic depletion of photoreceptors leads to a decrease in stomatal density in the corresponding spectra (Figure 6a) [176].

CONSTITUTIVE PHOTOMORPHOGENIC 1 (COP1), an E3 ubiquitin ligase, functions downstream of cryptochrome (CRY) and phytochrome (PHY) photoreceptors and acts as a key repressor of cell differentiation that negatively regulates stomatal development (Figure 6a). A series of experimental evidence shows that COP1-mediated regulation of photomorphogenesis takes place at the post-transcriptional level [180]. In the dark, COP1 activates the downstream YDA-MAPK cascade, suppressing stomatal cell lineage initiation, while active COP1 can lead to degradation of SCRM, suppressing stomatal formation in a YDA-independent manner [181]. Light inactivates COP1, which can no longer activate the YDA cascade, resulting in stomatal formation. Although the CRY signaling pathway is not fully understood yet, it has been found that under blue light conditions, CRY associates with COP1 [182], promoting cell differentiation processes that result in stomatal development. Moreover, it has been shown that cryptochromes and heterotrimeric G-proteins, whose role is well described in the regulation of many physiological processes, act antagonistically in the control of stomatal ontogenesis [183]. Although the underlying molecular mechanism is not fully understood, experimental data show that CRY1 positively regulates stomatal development by releasing the AGB1-mediated suppression of SPCH DNA-binding activity through the inhibition of AGB1-SPCH interaction to fine-tune stomatal density and patterning [183].

Another player in the light signal transduction pathway is the SUPPRESSOR OF PHYTOCHROME A1 (SPA1), a negative regulator of stomatal formation that associates with COP1 to repress cell differentiation, resulting in a reduced number of stomata (Figure 6a) [184,185]. In the presence of red light, SPA1 interacts with PHYB, resulting in the repression of the COP1-SPA1 complex, reinstating cell differentiation and the proliferation of MCs, resulting in increased stomatal formation [184,186]. Activation of PHYB also leads to the negative regulation of PHYTOCHROME INTERACTING FACTORS (PIFs), a class of cell differentiation repressors [184,186]. PHYB-mediated inhibition of the COP1-SPA1 complex leads to the translocation of COP1 from the nucleus to the cytoplasm. This translocation results in compartmentalized signaling cascades that trigger different outputs in response to light-quality signals [187,188]. The assembly of the PHYB-PIF4-SPA1 complex results in the accumulation of the light-responsive transcription factor HY5, which has a prominent role in photomorphogenesis, regulating essential developmental processes like chloroplast development, the formation of photosynthetic machinery, and pigment biosynthesis [189]. Far-red light activation of PHYA results in the autophosphorylation of the protein, which phosphorylates and inactivates PIFs [188].

An additional layer of regulation of stomatal development by light is the transcriptional induction of *STOMAGEN* [190]. Recently, HY5 was found to bind directly to the promoter of *STOMAGEN*, activating its expression in mesophyll cells [191,192]. It was recently reported that in *hy5* mutants, SPCH is hardly detectable, suggesting that HY5 is also required for the expression of the master transcription factor SPCH. This happens in a STOMAGEN-dependent manner, as HY5 directly activates *STOMAGEN* in mesophyll cells, which then stabilizes SPCH at the epidermis. Genetic inhibition of *STOMAGEN* or overexpression leads to stomatal formation that is unresponsive to light intensity [191]. The light-dependent mechanisms that regulate stomatal development are not yet fully understood, as light signaling intersects with hormones, temperature, and nutrients to control positive or negative stomatal development [193,194,195].

### 6.2. CO_2_ and Stomatal Development

Atmospheric levels of CO_2_ are continuously increasing due to the usage of fossil fuels for energy, contributing to the greenhouse effect. Research has shown that most plant species decrease stomatal density in response to increasing CO_2_, suggesting that CO_2_ negatively regulates stomatal formation [178,196] and, consequently, plant CO_2_ exchange and physiology [197]. It is therefore necessary to understand the plant molecular mechanisms regulating stomatal development under alterations of CO_2_ levels to predict how plants and ecosystems can adapt to increased CO_2_ levels.

At the molecular level, research to date has revealed two genes acting in plant responses to elevated CO_2_ levels: HIGH CARBON DIOXIDE (HIC), a 3-ketoacyl-coA synthase (KSC) that alters cuticular wax composition in the leaf epidermis, modulates CO_2_ diffusion gradients and the tissue absorption spectrum, resulting in changes in stomatal patterns and distribution (Figure 6d) [196,198]. Studies have demonstrated that high CO_2_ levels negatively regulate the EPF2 signal peptide, leading to repression of stomatal formation [199,200]. The molecular mechanism involves β-CARBONIC ANHYDRASE 1 and 2 (βCA1 and βCA2), which act together with CO_2_ RESPONSE SECRETED PROTEASE (CRSP), a protease localized in the extracellular space, to degrade EPF2 [199,200]. Furthermore, overexpression of the positive regulator of satellite stomatal formation, the DNA replication licensing factor CDC6, increased stomatal production in response to increased CO_2_ concentrations [197]. A recent study coupled the function of cyclin-dependent kinase 8 (CDK8), a mediator kinase involved in cell proliferation and differentiation, with the proper stomatal patterning and spacing under prevailing CO_2_ conditions [201]. CDK8 phosphorylates SPCH, regulating its protein levels and therefore the transcriptional activation of its transcriptional targets [201].

### 6.3. Stomatal Development and Temperature

Under elevated environmental temperatures, plants need to adjust their development to balance water conservation and evaporative cooling through the control of the rate of transpiration by regulating stomatal conductance. Along with the quick response of stomatal apertures to high temperatures, regulation of stomatal development in newly formed organs is an additional mechanism controlling plant body temperature. Heat stress, most of the time, is connected to water deficit. Under such conditions, plants with fewer stomata, such as *EPF1* overexpression lines, display lower and more stable leaf temperatures than plants with higher stomatal density, like *epf1epf2* mutants [85,173]. Interestingly, at high temperatures, *EPF1*-overexpressing plants showed larger stomatal apertures compared to wild-type plants, which balances the lower stomatal density (Figure 6e). This suggests that plants with restricted stomatal development show enhanced water conservation and more efficient regulation of stomatal function. It has been demonstrated that plants exposed to high temperatures develop fewer stomata [202]. This response relies on the function of PIF4, which is significantly upregulated by high temperatures [203]. Regarding stomatal development, PIF4 was found to bind to the promoter of *SPCH* to repress its expression, inhibiting the initiation of stomatal cell lineage [204]. Intriguingly, SPCH can also suppress *PIF4* expression through a negative feedback mechanism [204].

A recent study demonstrated the active role of HEAT SHOCK PROTEIN 90s (HSP90s) in the regulation of stomatal development under normal and heat stress conditions [205]. HSP90s associate with YDA, inhibiting the phosphorylation and activation of MPK3 and MPK6, leading to insufficient SPCH inactivation and, thus, an increased number of cells entering the stomatal cell lineage and the formation of stomatal clusters (Figure 6e) [205]. In the same study, HSP90s were found to regulate YDA polarity in stomatal lineage cells as well as SPCH transcriptional activity, changing the expression levels of several SPCH transcriptional targets, suggesting that HSP90s function as negative regulators of stomatal formation [205]. Furthermore, HSP90s also regulate stomatal differentiation under both normal and heat stress conditions [206].

### 6.4. Stomatal Development and Drought

Global climate change is intensifying water deficit-related problems, causing a severe reduction in crop yields. A major strategy that plants employ against drought is the reduction in stomatal conductance. Although this strategy supports plant survival, it does not favor high yields and productivity due to the reduction in photosynthesis. Several studies have shown that water deficit reduces stomatal density (Figure 6c) through the MAPK-dependent inactivation and destabilization of SPCH, resulting in the inhibition of stomatal cell lineage initiation [207]. ANGUSTIFOLIA3 (AN3) is a transcriptional regulator involved in stomatal development under drought conditions [208,209]. Highly induced YDA expression was reported in an3 mutants, while ChIP analysis showed that AN3 binds to the promoter of *YDA*, suggesting that AN3 acts as a transcriptional repressor of *YDA* [208,209].

Experiments in poplar showed that dehydration upregulates *PdEPF1*, resulting in lower stomatal density, higher water-use efficiency, and drought tolerance [209], making EPF1 a promising target for genetic modification. STOMATAL DENSITY AND DISTRIBUTION 1 (SDD1), a member of the subtilisin-like serine protease family that cleaves a mobile peptide negatively regulating stomatal development, seems to be another interesting target for genetic manipulation. Genetic inhibition of *SDD1* leads to increased stomatal densities and stomatal clustering, while the transcript levels of SDD1 severely increase in drought conditions [210]. Overexpression of *SchSDD1* in Arabidopsis and tomato improved plant drought tolerance and enhanced water-use efficiency, resulting in higher productivity [211]. The molecular pathway by which SSD1 suppresses stomatal development remains unclear, though. SDD1 expression is regulated by the GT2-LIKE 1 (GTL1) transcription factor, which binds to the *SDD1* promoter to repress its transcription [212], as in gtl1 mutants, SDD1 is induced and stomatal density is severely reduced, while *GTL1* expression is downregulated under drought conditions [212]. SDD1 expression is also controlled by the Ca^2+^-binding protein calmodulin, which in drought stress conditions binds to GTL1 and prevents its binding to the promoter of *SDD1*, de-repressing *SDD1* transcription [210].

Recent reports show that SPCH is the key point in the regulation of stomatal development, and this is the reason why many developmental signals and environmental stimuli target SPCH activity [213], while plants with lower levels of SPCH protein display enhanced drought tolerance [214]. INDETERMINATE DOMAIN 16 (IDD16), a C_2_H_2_ zinc-finger transcription factor, was recently characterized as a repressor of *SPCH* transcription through its binding to the promoter of *SPCH*. Genetic inhibition of *IDD16* results in higher stomatal density, while overexpression of *IDD16* reduces stomatal formation [214].

Recent studies reveal that in C3 plants, drought suppresses stomatal development by downregulating *SPCH* expression, whereas drought-tolerant C3 crops maintain normal SPCH levels and stomatal patterning [215,216]. In contrast, most C4 plants exhibit stable *SPCH* expression and undisturbed stomatal development under drought conditions. Intriguingly, drought-tolerant sugarcane diverges from this typical C4 response, instead displaying a C3-like repression of SPCH and stomatal development during water stress, unlike drought-susceptible C4 species. This distinction suggests divergent upstream regulatory mechanisms governing SPCH in drought-tolerant versus drought-susceptible C4 plants. By modulating stomatal development, drought-tolerant sugarcane minimizes water loss, thereby enhancing its resilience. Further investigation into the genomic and epigenetic basis of these differential SPCH regulatory pathways could provide critical insights into the evolution of key functional genes and the molecular foundations of drought tolerance.

### 6.5. Stomatal Development and ROS Signaling

During abiotic stresses, reactive oxygen species (ROS) are produced, such as superoxide radical, hydrogen peroxide (H_2_O_2_), or nitric oxide (NO), which are involved in defense mechanisms and act as signaling particles [217]. Recent findings show that spatially patterned H_2_O_2_ plays a critical role in stomatal development [218]. H_2_O_2_ accumulates in meristemoids due to the spatial expression of the H_2_O_2_ scavenging enzymes CAT2 and APX1. SCPH suppresses the expression of *CAT2* and *APX1* in meristemoids, while mutants of these genes display increased stomatal indices [217]. In addition, H_2_O_2_ presence activates the energy sensor SnRK1, which, through its KIN10 catalytic subunit, stabilizes SPCH, promoting stomatal lineage initiation (Figure 6b) [218].

In *Zea mays*, ROS function as crucial partners of auxin in establishing polarity and facilitating asymmetric divisions in SMCs [136]. H_2_O_2_ specifically accumulates at polarized SMC sites adjacent to GMCs, where it promotes SMC polarization and asymmetric division to form SCs. H_2_O_2_ simultaneously modulates auxin transport and distribution, counteracting the inhibitory effects of auxin transport inhibitors. Notably, H_2_O_2_ accumulation at polarized SMC sites correlates with localized auxin accumulation in both the SMCs and neighboring cells. Disruptions in ROS homeostasis impair SMC polarization and division, while simultaneously negating auxin’s positive regulatory effects on SC formation [136].

## 7. Conclusions and Future Perspectives

Plants, as sessile organisms, needed to develop unique and sophisticated mechanisms that permitted their survival under adverse conditions, often in marginal environments. Stomata, specialized epidermal structures unique to plants, serve as critical hubs for perceiving and responding to diverse environmental signals. These microscopic pores, functioning as primary sensors, play a pivotal role in balancing the fundamental trade-off between plant growth and stress defense. Although the stomata’s role in abiotic stress response is established, the molecular basis underlying this function is under active investigation.

Recent advances in cutting-edge imaging technologies and novel analytical tools have enabled significant progress in deciphering the complex regulatory networks controlling stomatal lineage progression. These developments are providing unprecedented insights into how both genetic and environmental factors influence cell fate decisions and division patterns in stomatal development. In the present review, we summarize recent findings on stomatal development under abiotic stresses, focusing on how stress-induced modulation of core transcriptional regulators in the stomatal signaling pathway, such as SPCH, MUTE, and FAMA, drives morphological changes in stomatal patterning and structure across the leaf epidermis.

A key challenge in modern plant biology, which also applies to research on stress-responsive stomatal development, is the possibility of translating and transferring the obtained knowledge from model plants to other economically important species. Given that the majority of research is conducted in Arabidopsis, the model plant for dicots, and the fact that stomatal development between dicots and monocots differs in many aspects, there is a great demand to expand research in monocot species. Hopefully, potential discoveries about new molecular mechanisms guiding stomatal responses to stress will benefit the agricultural sector through the generation of new, more resistant plant varieties in the context of global climate change. In addition, future research focusing on understudied basal lineages (such as hornworts, ancient lycophytes, and early diverging monocots), combined with functional analyses of stomatal genes across the phylogeny, will further illuminate the deep evolutionary origins and diverse trajectories of this critical plant structure.

## Figures and Tables

**Figure 1 plants-14-02405-f001:**
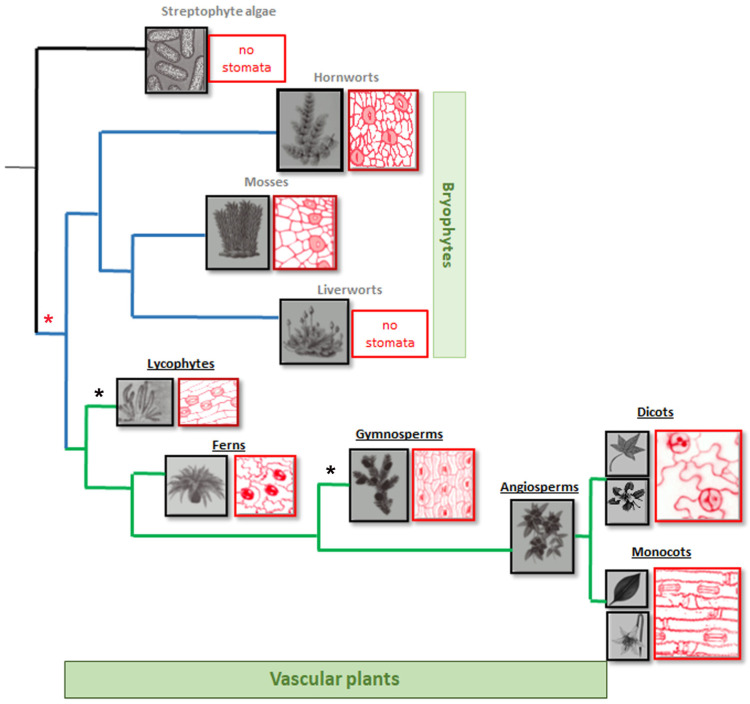
The phylogenetic context of stomatal diversity across Land Plant Taxa. Land plants are divided into vascular (green lines) and nonvascular (blue lines) groups. Epidermal tracings represent the wide diversity of stomatal complexes among these groups. The red star indicates the appearance of stomata. The black star marks the acquisition of active stomatal control.

**Figure 2 plants-14-02405-f002:**
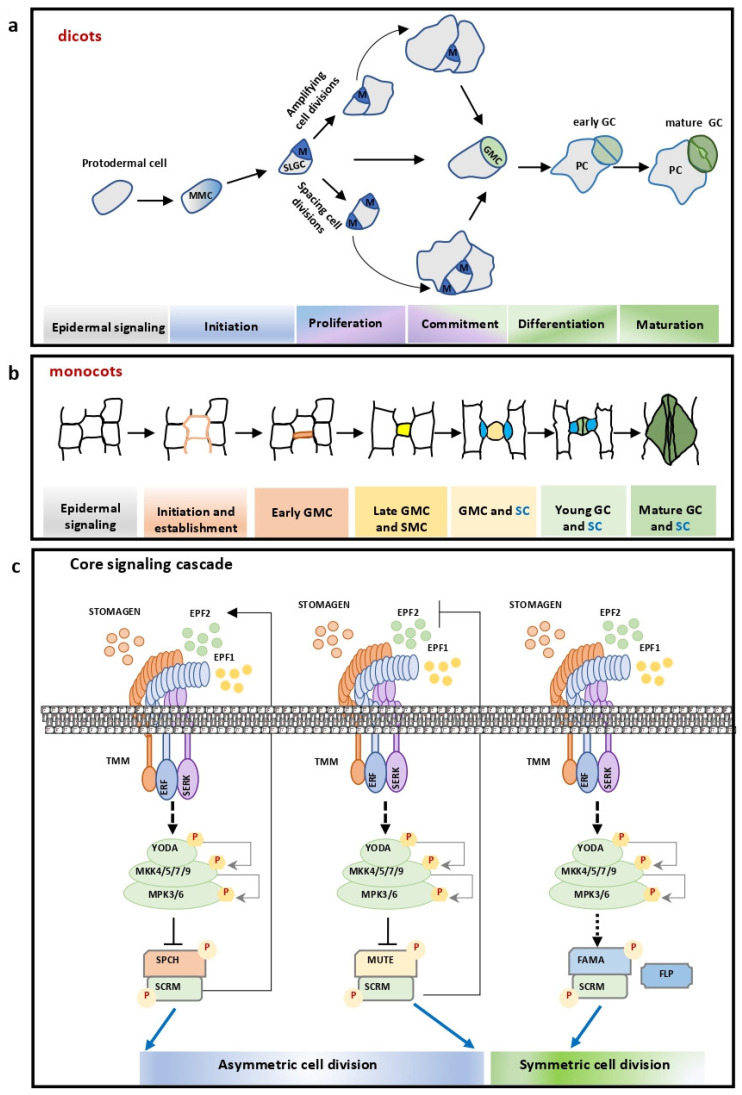
**A model of the stomatal lineage in dicotyledonous and monocotyledonous plants.** (**a**) Schematic representation of cell divisions in *Arabidopsis thaliana* (eudicot). A protodermal cell, after receiving the appropriate stimuli, initiates the stomatal lineage, becoming a meristemoid mother cell (MMC). MMCs undergo asymmetric divisions, resulting in a larger daughter cell, a stomatal lineage ground cell (SLGC), and a smaller meristemoid (M). Meristemoids then differentiate into a guard mother cell (GMC) or undergo further asymmetric cell division (amplifying) to complete stomatal development. Amplifying divisions serve to increase the number of stomatal lineage cells, both by regenerating the meristemoid itself and by producing larger daughter cells that can contribute to the overall epidermal cell population. The GMC divides symmetrically to give rise to a mature guard cell pair (GC) surrounding the stomatal pore. The SLGC differentiates into pavement cells or adopts an MMC fate and undergoes spacing divisions (asymmetric) to generate secondary meristemoids (Ms). The spacing divisions are oriented following the “one-cell spacing” rule so that the two stomata are spaced by at least one cell. (**b**) Schematic representation of stomatal lineage in monocotyledonous plants. Stomatal development starts with protodermal signaling that leads to the selection of stomatal lineage cells, which then divide asymmetrically to generate guard mother cells (GMCs). The initial rectangular GMCs expand and become square, inducing the polarization of subsidiary mother cells (SMCs). The SMCs divide asymmetrically to generate two SCs. The maturation of SCs ensues while GMCs divide symmetrically to produce two guard cells. Subsequently, elongation and maturation of GCs take place. (**c**). The core signaling cascade of stomatal development. The core signaling cascade starts with receptor–ligand interactions. TMM associates with ER family (ERf) receptor kinases in the cytoplasm. Stomatal divisions are regulated by EPF1/2 peptides, which bind to the ERf/TMM/SERK receptor complex to activate it. STOMAGEN competes with EPF1/2 for the binding sites of the ERf/TMM/SERK complex. Receptor activation triggers the downstream YDA MAPK signaling cascade and thereby activates MKK4/5 or MKK7/9 and then MPK3/6. The YDA MAPK signaling cascade interacts with the master transcription factors regulating stomatal lineage: SPEECHLESS (SPCH), MUTE, and FAMA. SPCH/SCRM regulates the entry and amplifying divisions, being responsible for lineage specification and proliferation. MUTE/SCRM directs the development from the meristemoid to the GMC, being responsible for fate commitment, and FAMA/SCRM regulates guard cell division, as well as differentiation and maturation. Arrows and bar-ended lines show activation and inhibition, respectively. Dotted lines indicate indirect regulation.

**Figure 3 plants-14-02405-f003:**
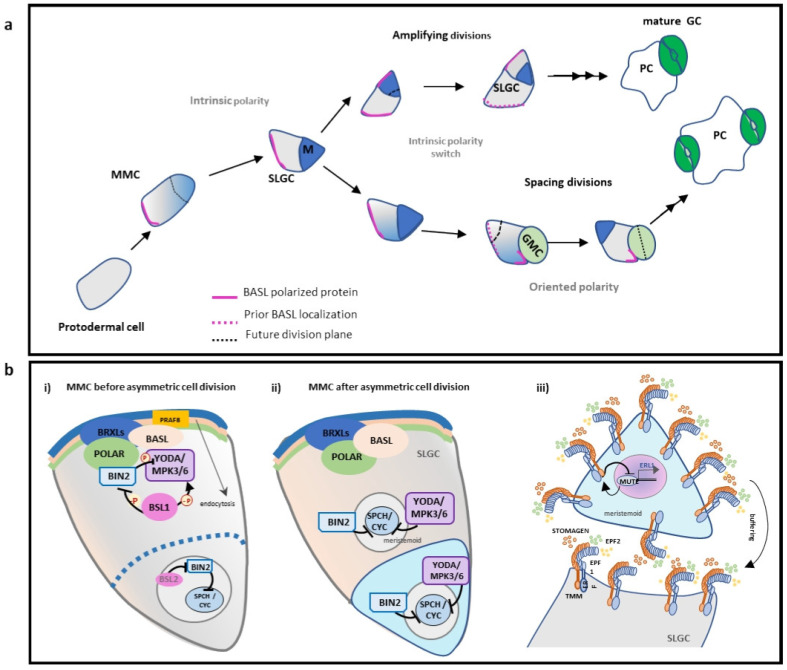
**Asymmetric cell divisions, division polarity, and assembly of polarity components during stomatal development in Arabidopsis.** (**a**) Stomatal lineage neighbor cells affect the asymmetric divisions of MMCs. Schematic representation of polarity, which influences cell sizes and division orientation and is marked by the localization of the cortical proteins BASL and POLAR (pink lines). Before the asymmetric division of MMCs, BASL localizes to the cortex of the MMC in crescents at the side where the future SGLC will be formed. BASL localization creates cell polarity, which acts as a repellant of the nucleus and the following division plane. In amplifying divisions self-renewing meristemoids reorient BASL/POLAR cortical crescents away from the most recently formed cell walls, preventing the formation of adjacent stomata in the lineage. Black dotted lines indicate the position of the future division plane. In SLGCs that have adopted an MMC fate, BASL/POLAR should redistribute within the same cell, creating a crescent at the pre-existing flanking meristemoid or GMC. (**b**) Schematic representation of the polarity complex assembly during asymmetric cell division (**i**,**ii**). The formation of the preprophase band indicates the entry of the cell into mitosis (dotted line). The nucleus migrates away from the polarity site before division and towards the polarity site after division. (**i**) Before asymmetric cell division in an MMC, BASL (salmon) is polarized at the cell cortex through interaction with BRX that is palmotoylated to attach to the plasma membrane (blue). PRAF/RLD, plant-specific proteins, physically interact with the BASL anchor at the plasma membrane and determine the polarization of the protein and stomatal asymmetric cell division. The polarization of POLAR also depends on BASL. Polarized POLAR associates with BIN2 GSK3-like kinase (light blue) and inhibits the kinase activity of YDA (pink), resulting in the alleviation of MAPK-mediated suppression of SPCH (blueish). High levels of SPCH increase the cell division potential of MMCs. (**ii**) During asymmetric division, BSL1 (intense pink) is associated with the BASL polarity complex. The polarization of BSL1 coincides with preprophase band formation, which marks the entry of the MMC into mitosis. Polarized BSL1 phosphatase suppresses BIN2 activity at the plasma membrane, leading to the dissociation of BIN2 from the polarity complex. Therefore, YDA inhibition by BIN2 is alleviated, resulting in the inactivation and degradation of SPCH. At the plasma membrane, BIN2, through phosphorylation, induces the turnover of POLAR, resulting in lower levels of membrane-bound BIN2. Increased levels of BIN2 at the nucleus further promote the degradation of SPCH. After asymmetric cell division, the BSL1-BASL-YDA polarity complex is inherited by SLGC, and subsequently, BSL1 activates the YDA cascade through phosphorylation. Elevated MAPK signaling strongly suppresses SPCH, leading to lower cell division potential. Timed assembly of the polarity complex is essential for the progression of stomatal asymmetric cell division and cell fate specification of the two daughter cells that will follow distinct developmental trajectories. (**iii**) The stomatal differentiation potential is also regulated by EPF1-ERL1 ligand–receptor signaling. Both ERL1 and EPF1 are expressed in meristemoids and SLGCs since they were both upregulated by SPCH in earlier stages. During the transition of the meristemoid to the guard mother cell (GMC), the transcription factor MUTE upregulates the expression of ERL1, causing accumulation of ERL1 in meristemoids compared to the levels of ERL1 in SLGCs. EPF1 is expressed and secreted in late meristemoids. EPF1 binding to ERL1 and TMM induces the downregulation of MUTE. EPF1, also perceived by ERL1 and TMM, expressed in neighboring SLGCs, triggers downstream signal transduction to ensure asymmetric spacing division. Therefore, receptors acting in meristemoids and SLGCs buffer the pools of EPF1 ligand peptides. Arrows and bar-ended lines show activation and inhibition, respectively.

**Figure 4 plants-14-02405-f004:**
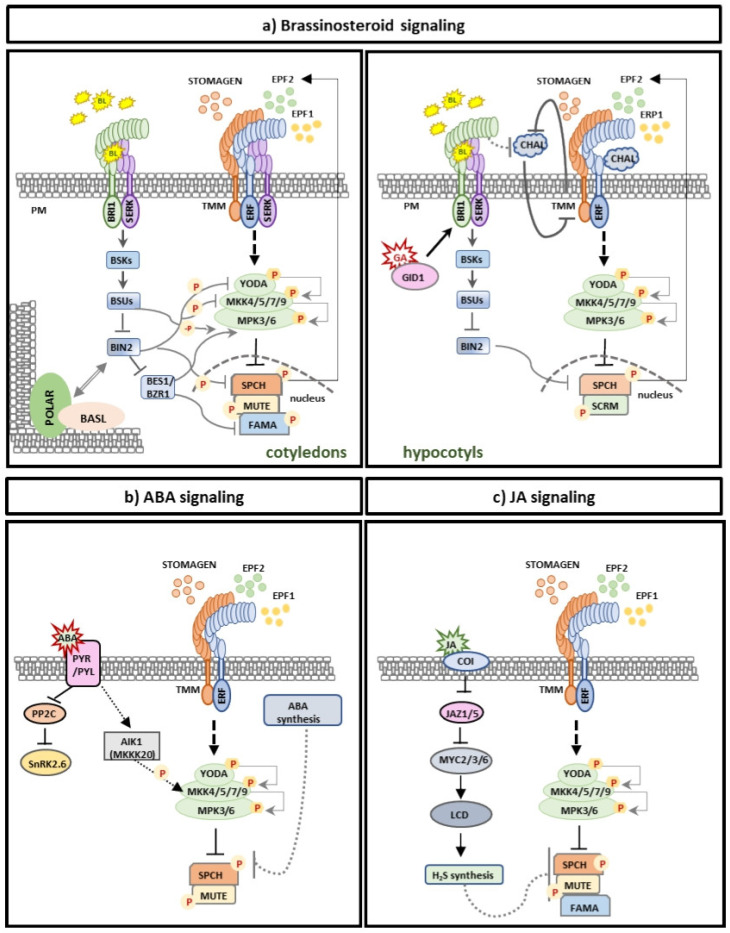
**Integration of BR, GA, ABA, and JA signaling pathways into the core stomatal signaling cascades**. (**a**) Brassinosteroids (BRs) suppress stomatal development at the cotyledons. Under low concentrations of BRs, POLAR recruits BIN2, the negative regulator of BR signaling, at the cell cortex, where the major polarity protein BASL localizes. This results in BIN2-mediated attenuation of the YDA MAPK cascade and the release of BIN2-mediated inhibition of SPCH in the nucleus, promoting asymmetric cell divisions. In the presence of BRs, BIN2 inactivation leads to the dissociation of BIN2 from the BASAL-POLAR complex, inhibiting asymmetric cell division. The activation of the YDA MAPK signaling cascade results in the inactivation and degradation of SPCH, repressing stomatal development. BZR1 and BES1 positively regulate the expression of MKK9 to repress FAMA expression in the cotyledons of etiolated seedlings. The induced expression of MKK9 inhibits the transition of meristemoids to GMCs. Decreased levels of FAMA block the differentiation from GMCs into mature guard cells. BRs promote stomatal formation in hypocotyls. Stomatal formation in hypocotyls does not involve the YDA MAPK signaling cascade. Stomatal production in hypocotyl relies on the BR-mediated inactivation of BIN2 that alleviates SPCH and ICE1. Gibberellins (GAs) promote stomatal formation in hypocotyls, too. Although the GAs’ mode of action is unclear, experimental data support that they crosstalk with the BR-mediated promotion of stomatal formation in hypocotyls. Activation of GID1, the GA receptor, results in the upregulation of the BRI1 receptor, thus activating the BR signaling cascade. (**b**) Abscisic acid (ABA) inhibits entry into the stomatal cell lineage. This inhibition is based on the transcriptional suppression of master transcription factors of stomatal development, such as SPCH and MUTE. (**c**) Jasmonic acid (JA) suppresses stomatal formation and distribution through inhibition of SPCH, MUTE, and FAMA expression by the JA master regulators MYC2/3/4. MYC2/3/4 promote the expression of the H2S biosynthesis gene LCD. The way that H2S suppresses SPCH/MUTE and FAMA remains elusive. Arrows and bar-ended lines show activation and inhibition, respectively. Dotted lines indicate indirect regulation.

**Figure 5 plants-14-02405-f005:**
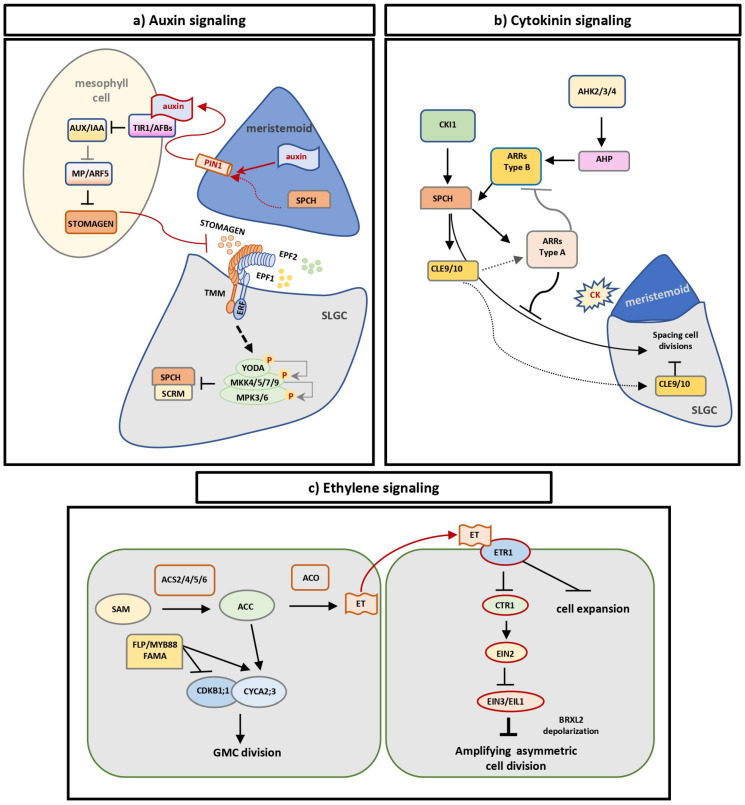
**Integration of auxin, cytokinin, and ethylene signaling pathways into the core stomatal signaling cascades to regulate stomatal development and patterning.** (**a**) Auxin signaling, transport, and biosynthesis participate in the control of stomatal development. The highest auxin activity is observed in early meristemoids and decreases during the transition to GMCs. The release of auxin from GMCs facilitates the symmetric division of GMCs. Auxin signaling leads to the activation of MP/ARF5, which directly suppresses the transcription of STOMAGEN in mesophyll cells and, in turn, results in the de-repression of the ERf/TMM receptors and the subsequent activation of the YDA-MAPK signaling cascade that suppresses stomatal initiation. (**b**) Cytokinin (CK) promotes spacing asymmetric divisions, enhancing stomatal development. CK signaling leads to the activation of SPCH, which induces the transcription of ARR16/17 and CLE9/10 at the stomatal lineage, constituting a negative regulatory loop that controls the SLGC spacing divisions. (**c**) Ethylene (ET) acts as a regulator of stem cell and GMC divisions during stomatal development. Amplifying cell divisions are repressed by ET signaling through the depolarization of BRXL2. The precursor of ET, ACC, promotes the symmetric cell division of GMCs and competes with the transcriptional activity of FAMA and FLP/MYB88 on CYDKB1;1/CYCA2,3. Arrows and bar-ended lines show activation and inhibition, respectively. Dotted lines indicate indirect regulation.

**Figure 6 plants-14-02405-f006:**
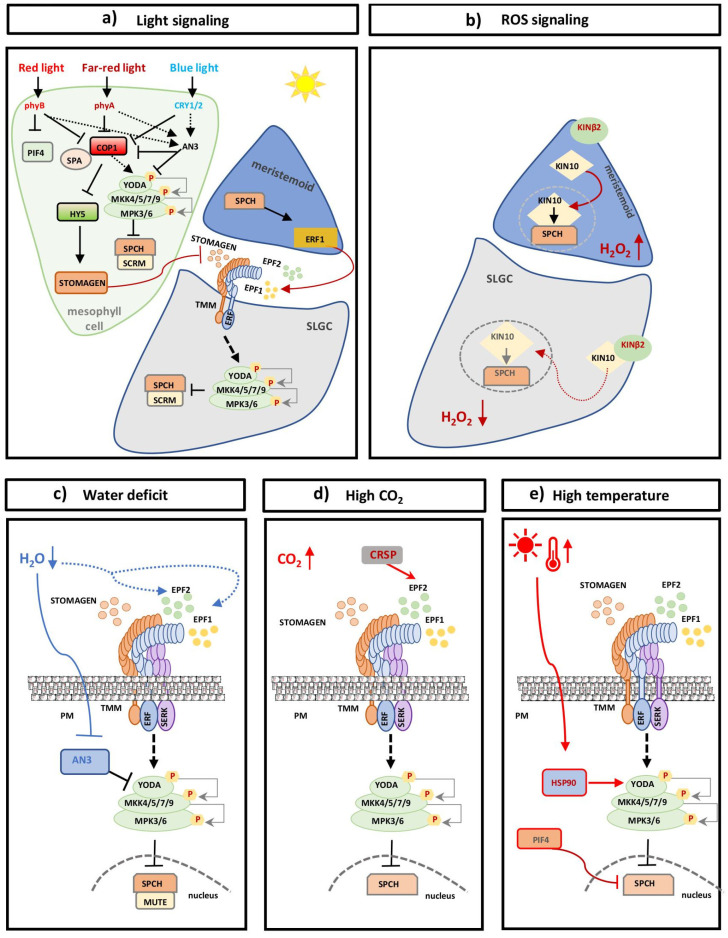
**The impact of environmental stimuli on the control of stomatal formation. Schematic depiction of stomatal development control by systemic signaling.** (**a**) In the presence of intense light conditions, stomatal formation is induced. Specifically, blue light activates CRY, which promotes stomatal development. CRY inhibits COP1, an inhibitor of stomatal development, resulting in the accumulation of HY5. COP1 activates the YDA-MAPK signaling cascade, resulting in the degradation of SPCH and ICE1. The detailed mechanism of COP1-mediated activation of the YDA cascade remains elusive. In mesophyll cells HY5 binds to the promoter of STOMAGEN, activating its transcription. Elevated STOMAGEN levels in turn inhibit the repressive action of ERf epidermal signaling, leading to the accumulation of SPCH, which mediates the entry of cells into the stomatal lineage and, consequently, stomatal production. The formation of the COP1-SPA complex inhibits the accumulation of HY5. Far-red light also activates PHYA, which promotes stomatal production by suppressing PIF transcription factors that act as inhibitors of stomatal formation. Red light activates PHYB, which also induces stomatal development. PHYB inhibits PIFs and the formation of the COP1-SPA complex, inhibitors of stomatal development. Light also promotes the expression of AN3, most likely through photoreceptor-mediated pathways to repress the expression of COP1 and YDA. (**b**) Spatially patterned H_2_O_2_ in epidermal cells is critical for optimal stomatal formation. SPCH directly binds and represses the expression of CAT2 and APX1, H_2_O_2_ scavenging enzymes, leading to increased levels of H_2_O_2_ in meristemoids. The elevated levels of H_2_O_2_ repress the association between KIN10 and KINβ2 and promote the nuclear translocation of KIN10, thereby stabilizing SPCH to induce stomatal lineage initiation. (**c**) Water deficit results in the suppression of stomatal formation. Water unavailability represses AN3, a negative regulator of the YDA-MAPK signaling cascade, leading to increased levels of SPCH. At the same time, water deficit enhances EPF1/2 signaling, activating the ERf receptors that trigger the activity of the YDA-MAPK signaling cascade to suppress the expression of the MUTE transcription factor. (**d**) Under high concentrations of CO2, stomatal development is suppressed through an increase in HIC levels, an inhibitor of stomatal formation. High levels of CO_2_ also induce the expression of CRSP and βCA, which form a complex that inhibits stomatal formation by repressing the expression of the EPF2 signal peptide, which promotes stomatal development (**e**). Heat stress suppresses stomatal development. Under heat stress conditions there is an increase in HSP90 levels. HSP90s interact with YDA and mediate activation through phosphorylation of MPK6 and MPK3, leading to the phosphorylation and inactivation of SPCH. At the same time, heat stress induces the expression of PIF4, an inhibitor of stomatal production that binds specifically to the promoter of SPCH, inhibiting its transcription. Arrows and bar-ended lines show activation and inhibition, respectively. Dotted lines indicate indirect regulation.

## Data Availability

No data was used for the research described in the article.

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
