# Peer review of "Guardians of Water and Gas Exchange: Adaptive Dynamics of Stomatal Development and Patterning"

_plants, 2025, doi:10.3390/plants14152405_

Round 1
Reviewer 1 Report
Comments and Suggestions for Authors
The manuscript represents a broad review of stomata biogenesis and functionality. The review covers a wide spectrum of stomata related aspects from ancient evolution to interaction with various environmental factors. Representing a summary of more than 200 publications, the review could be of interest for many scientists working in different areas of plant science. However, before the manuscript could be accepted for publication, some issues should be addressed for improvement and clarification.
In general, some chapters are too long and contain excessive description of items not directly related to the main topic. This should be seriously addressed, especially chapter 5.1, should be definitely revised and shorten. There are a number of too long sentences, confusing and/or difficult to understand.
Specific comments:
Citations 76 and 83 refer to the same publication;
Lines 59-61: the meaning of the statement is not quite clear;
Line 263: Apparently, ER states for endoplasmic reticulum, should be spelled out when first time mentioned.
Line 361: the use of term ”pharmacological” does not look appropriate in the context;
Line 417-418: Not a quite correct statement.
Line 436-440: Not clear how this statement relates to BR;
Line 476: I would recommend to replace word “supported” with “suggested”.
Line 523: The sentence talks about “three groups of finding”, not clear what they are. The statement/paragraph should be restructured/rephrased.
Line 573: “Moreover, SPCH functions as an effector of cytokinin signaling, inducing the expression of ARABIDOPSIS (Figure 573 5b).” What “ARABIDOPSIS” ?
Line 599: “ACC” should be spelled out;
Lines 681-686: The two sentences are too long, over complicated, not easy to comprehend. Please make them shorter and more to the point.
Author Response
Reviewer1
Comments and Suggestions for Authors
The manuscript represents a broad review of stomata biogenesis and functionality. The review covers a wide spectrum of stomata related aspects from ancient evolution to interaction with various environmental factors. Representing a summary of more than 200 publications, the review could be of interest for many scientists working in different areas of plant science. However, before the manuscript could be accepted for publication, some issues should be addressed for improvement and clarification.
In general, some chapters are too long and contain excessive description of items not directly related to the main topic. This should be seriously addressed, especially chapter 5.1, should be revised and shorten. There are a number of too long sentences, confusing and/or difficult to understand.
Answer: We thank the reviewer for the overall positive and valuable comments that helped us to improve the manuscript. In general, we carefully revised the whole manuscript, and we tried to address all the comments. We revised the chapter 5.1 making it shorter and we rephrased most of the long sentences.
Specific comments:
- Citations 76 and 83 refer to the same publication;
Answer: We thank the reviewer for the comment. We have revised the references in the whole manuscript.
- Lines 59-61: the meaning of the statement is not quite clear;
Answer: We thank the reviewer for the comment. We have revised this part of the manuscript, which goes as follows: “Previous studies have shown that water balance in guard cells took place about 360 million years ago [7], while the emergence of an abscisic (ABA_-dependent stomatal regulation mechanism of stomatal pore closure seems to have emerged 400 million years ago [8]. The discovery of genetic variation in periods of climate succession is an approach for comprehending the mechanisms governing the kinetics of stomatal opening and closure as an adaptation mechanism to unfavorable environments.”
- Line 263: Apparently, ER states for endoplasmic reticulum, should be spelled out when first time mentioned.
Answer: ER is the abbreviation of ERECTA receptors, which have been mentioned above.
- Line 361: the use of term ”pharmacological” does not look appropriate in the context;
Answer: The term "pharmacological has been removed from the revised version
- Line 417-418: Not a quite correct statement.
Answer: We thank the reviewer for the comment. We revised the sentence and now it goes as follows: “Answer, as stomata regulate photosynthesis while optimizing water use efficiency (WUE)”.
- Line 436-440: Not clear how this statement relates to BR;
Answer: We thank the reviewer for the comment. In this part of the manuscript, which is the following: “Recently, EPF1, EPF2, and EPFL9/STOMAGEN were found to regulate cellular divisions and cell fate transitions [115]. This binding promotes the association of ER-family/TMM receptor complex with SOMATIC EMBRYOGENESIS RECEPTOR KINASE 1, SERK2, SERK3 (also known as BAK1), and SERK4 to convey the cellular signal [75, 116, 117, 118]. Notably, only the higher-order serk mutants exhibit severe stomatal clustering, while SERKs have been shown to associate with ER/ERLs in a ligand-induced mode [117].”
we refer to BAK1, the coreceptor of BRI1 in the BR signaling cascade, and its role in the stomatal development core cascade.
- Line 476: I would recommend to replace word “supported” with “suggested”.
Answer: We replaced supported with suggested
- Line 523: The sentence talks about “three groups of finding”, not clear what they are. The statement/paragraph should be restructured/rephrased.
Answer: We rephrased the sentence in the revised version of the manuscript and now it goes as “This notion relies on the following findings:….”
- Line 573: “Moreover, SPCH functions as an effector of cytokinin signaling, inducing the expression of ARABIDOPSIS (Figure 573 5b).” What “ARABIDOPSIS” ?
Answer: The sentence has changed in the revised version, and it goes as follows: “Moreover, SPCH functions as an effector of cytokinin signaling, inducing the expression of type-A ARR16 (Figure 5b)”.
- Line 599: “ACC” should be spelled out;
Answer: We thank the reviewer for the comment. We have spelled out ACC in the revised manuscript
- Lines 681-686: The two sentences are too long, over complicated, not easy to comprehend. Please make them shorter and more to the point.
Answer: We thank the reviewer for the comment. We have revised this part using shorter sentences. The revised part is the following: “Environmental signals influence diverse physiological processes that govern plant development. As stomata are critical for key functions such as photosynthesis and nutrient uptake, their development is tightly regulated by environmental cues [171] to optimize the balance between photosynthetic efficiency and transpiration under varying conditions [172]. Key factors modulating stomatal ontogenesis—including stomatal density, size, and index—are light intensity, water availability, temperature, and atmospheric COâ‚‚ concentration [173, 174].”

Reviewer 2 Report
Comments and Suggestions for Authors
General comments:
This work ‘’ Guardians of Water & Gas Exchange: Adaptive Dynamics of Stomatal Development and Patterning’’ presents an interesting and relevant topic, aims to provide an understanding of the complex interplay between stress stimuli and key factors regulating stomatal development. The study is well structured, however, some improvements are needed. In addition, the m/s does not respect the plant journal instructions.
Specific comments
the title is precise, informative, and reflects the content and scope of the review
Keywords: please add more the words and delete the numbers
Please add missing references: from line 31 to line 44, no references were cited....
It is recommended to insert the study objectives at the end of the introduction
Check that all gene names are in italics.
L421-L422: Please provide the full form of the abbreviation 'BR' and avoid starting the paragraph with an abbreviation.
L545: Check that all species names are in italics.
Author Response
Reviewer 2
General comments:
This work ‘’ Guardians of Water & Gas Exchange: Adaptive Dynamics of Stomatal Development and Patterning’’ presents an interesting and relevant topic, aims to provide an understanding of the complex interplay between stress stimuli and key factors regulating stomatal development. The study is well structured, however, some improvements are needed. In addition, the m/s does not respect the plant journal instructions.
Answer: We would like to thank the reviewer for the overall positive and constructive comments, which helped us to improve the manuscript. We believe that in the revised manuscript we have addressed point by point all the comments.
Specific comments
the title is precise, informative, and reflects the content and scope of the review
- Keywords: please add more the words and delete the numbers
Answer: We thank the reviewer for the suggestion. In the revised manuscript we added more keywords “stomatal development; stomatal differentiation; stomatal lineage; stomatal patterning; hormonal control; abiotic stress; “
- Please add missing references: from line 31 to line 44, no references were cited....
Answer: We thank the reviewer for the comment. The missing references have been added in the revised version of the manuscript.
- It is recommended to insert the study objectives at the end of the introduction
Answer: We thank the reviewer for the suggestion. In the revised manuscript we have added at the end of the introduction the objectives of the current review
- Check that all gene names are in italics.
Answer: We appreciate the valuable comment of the reviewer. We have made the appropriate changes.
5 L421-L422: Please provide the full form of the abbreviation 'BR' and avoid starting the paragraph with an abbreviation.
Answer: We thank the reviewer for the comment. We have made the change.
6 L545: Check that all species names are in italics.
Answer: We appreciate the valuable comment of the reviewer. We have made the appropriate changes.

Reviewer 3 Report
Comments and Suggestions for Authors
review of Plants-3759305
Guardians of Water & Gas Exchange: Adaptive Dynamics of Stomatal Development and Patterning
Eleni Giannoutsou, Ioannis-Dimosthenis Adamakis and Despina Samakovli
The authors reviewed the current status of our knowledge of stomatal development and how it is regulated in response to environmental cues. They have done a thorough, comprehensive review of the various factors controlling stomatal development and the evolution of stomata in land plants. The figures provide useful visual representations of their topics.
I would like to see the authors emphasize in the introduction that plants need to take up CO2 via stomates for photosynthesis, so the key tradeoff is how to optimize CO2 uptake while minimizing water loss.
Please be careful to italicize Latin binomial names throughout the paper
Please be careful to check throughout the paper for places where “stomata” should be “stomatal”
Line 23: Please change “one’ to “on”
Line 38: Please change “at’ to “in”
Lines 57-58: please rewrite for clarity
Section 3: please consider adding a sentence explaining that much of our knowledge of stomatal development comes from studies in Arabidopsis using genetic approaches.
Line 151: please change to” The process of stomatal development in grasses”
Line 152: please delete “of”
Lines 152- 160: please reorganize and rewrite for clarity
Line 166: please clarify “(amplifying)”
Line 173: please change to “Subsequently, elongation and maturation of GCs takes place.”
Line 210: please change MY88 to MYB88.
Lines 263-264: please rewrite for clarity
Lines 430-433: please rewrite to indicate that inactivation of BIN2 allows the transcription of BL-responsive genes since it is no longer inhibiting their expression.
Line 472: please change to “…spatiotemporal deactivation of the MAPK signaling cascade…”
Line 620: please change to “GA is a growth-promoting hormone…”
Lines 694-696: please rewrite for clarity
Author Response
Reviewer 3
review of Plants-3759305
Guardians of Water & Gas Exchange: Adaptive Dynamics of Stomatal Development and Patterning
Eleni Giannoutsou, Ioannis-Dimosthenis Adamakis and Despina Samakovli
The authors reviewed the current status of our knowledge of stomatal development and how it is regulated in response to environmental cues. They have done a thorough, comprehensive review of the various factors controlling stomatal development and the evolution of stomata in land plants. The figures provide useful visual representations of their topics.
- I would like to see the authors emphasize in the introduction that plants need to take up CO2 via stomates for photosynthesis, so the key tradeoff is how to optimize CO2 uptake while minimizing water loss.
Answer: We thank the reviewer for this comment. We have revised this part, which in the new version of the manuscript goes as follows: “Stomatal characteristics exhibit the strongest influence on gross primary productivity variation among other leaf anatomical traits. The former is based on the fact that even though stomata occupy just 1-3% of the leaf surface, they mediate over 95% of terrestrial COâ‚‚ uptake and nearly all water loss in plants. By regulating gas exchange, they act as the primary gatekeepers of photosynthetic efficiency and transpiration rates, making them critical determinants of water-use efficiency (WUE) in crops. In terrestrial plants, stomata play a pivotal role in transpiration, directly regulating hydraulic conductance through vascular tissues. This functional interdependence strongly supports the co-evolution of stomata with vascular systems [5].”
2 Please be careful to italicize Latin binomial names throughout the paper
Answer: We appreciate the valuable comment of the reviewer. We have made the appropriate changes.
3 Please be careful to check throughout the paper for places where “stomata” should be “stomatal”
Answer: We thank the reviewer for the comment. We checked the whole manuscript about the correct use of stomata and stomatal.
4 Line 23: Please change “one’ to “on”
Answer: We thank the reviewer for the comment. We made the change.
5 Line 38: Please change “at’ to “in”
Answer: We thank the reviewer for the comment. We made the change.
6 Lines 57-58: please rewrite for clarity
Answer: We thank the reviewer for the comment. This part has been changed as follows: “Previous studies have shown that water balance in guard cells took place about 360 million years ago [7], while the emergence of an abscisic (ABA_-dependent stomatal regulation mechanism of stomatal pore closure seems to have emerged 400 million years ago [8]. The discovery of genetic variation in periods of climate succession is an approach for comprehending the mechanisms governing the kinetics of stomatal opening and closure as an adaptation mechanism to unfavorable environments.”
7 Section 3: please consider adding a sentence explaining that much of our knowledge of stomatal development comes from studies in Arabidopsis using genetic approaches.
Answer: We thank the reviewer for the comment. This part has been changed as follows: “Stomatal development, which has been thoroughly studied in the model plant Arabidopsis thaliana using genetic approaches, proceeds through a tightly regulated sequence of asymmetric cell divisions and fate transitions, orchestrated by intercellular signaling pathways and cell-to-cell communication.”
8 Line 151: please change to” The process of stomatal development in grasses”
Answer: We thank the reviewer for the comment. This part has been changed in the revised manuscript.
9 Line 152: please delete “of”
Answer: We thank the reviewer for the comment. This part has been changed in the revised manuscript.
10 Lines 152- 160: please reorganize and rewrite for clarity
Answer: We thank the reviewer for the comment. This part has been revised as follows:
“Stomatal development in grasses follows a well-defined sequence of stages: a) establishment of the stomatal row, b) generation of the guard cell mother cell (GMC) via an initial asymmetric division, c) establishment of polarity in the subsidiary mother cell (SMC), d) generation of two subsidiary cells (SC) via two asymmetric divisions of the adjacent SMCs, e) formation of two immature guard cells through a symmetric division, and finally f) maturation of two dumbbell-shaped GCs [29,32, 33, 34, 35, 36]. During stomatal ontogenesis in grasses, young GMCs first appear near the base, as rectangular cells, before transitioning into a square morphology. At this stage, they induce neighboring cells to differentiate into SMCs. A microtubule (MT)-dependent mechanism subsequently reduces GMC width, triggering localized protrusion of adjacent SMCs toward the developing GMCs [33, 37, 38, 39, 40, 41]. Prior to asymmetric division, a cascade of polarization events ensures proper SC formation [30].”
11 Line 166: please clarify “(amplifying)”
Answer: We thank the reviewer for the comment. In the revised manuscript we explained better the term “amplifying”. The revise text goes as follows: “Amplifying divisions serve to increase the number of stomatal lineage cells, both by regenerating the meristemoid itself and by producing larger daughter cells that can contribute to the overall epidermal cell population.”
12 Line 173: please change to “Subsequently, elongation and maturation of GCs takes place.”
Answer: We thank the reviewer for the suggestion. We made the change.
13 Line 210: please change MY88 to MYB88.
Answer: We thank the reviewer. We have made the change.
14 Lines 263-264: please rewrite for clarity
Answer: We thank the reviewer for the suggestion. In the revised version of the manuscript we have rewritten the sentence, which is the following:
“The role of ER receptors in stomatal formation is well-established, however, the way that the activated ER receptors are desensitized and recycled is yet not well understood”.
15 Lines 430-433: please rewrite to indicate that inactivation of BIN2 allows the transcription of BL-responsive genes since it is no longer inhibiting their expression.
Answer: We thank the reviewer for the suggestion. We have revised the sentence which is the following: “In the presence of BL, BIN2 is inactive and can no longer phosphorylate and inactivate the master transcription factors BRI1-EMS-SUPPRESSOR1 (BES1) and BRASSINAZOLE RESISTANT 1 (BZR1). The active BES1 and BZR1, therefore, can induce the transcription of genes affected by BL signaling that regulate a broad spectrum of plant developmental processes [115].”
16 Line 472: please change to “…spatiotemporal deactivation of the MAPK signaling cascade…”
Answer: We thank the reviewer for the comment. In the revised manuscript, the sentence has changed, and it is as follows: “These findings highlight the essential role of spatiotemporal deactivation of the MAPK signaling cascade in stomatal cell lineage determination, which is mediated by the differential function of BSL phosphatases [123]”.
17 Line 620: please change to “GA is a growth-promoting hormone…”
Answer: We thank the reviewer for the comment. In the revised version of the manuscript, we have made the change.
18 Lines 694-696: please rewrite for clarity
Answer: We appreciate the comment of the reviewer and we revised the sentence as follows: “CONSTITUTIVE PHOTOMORPHOGENIC 1 (COP1), an E3 ubiquitin ligase, functions downstream of cryptochrome (CRY) and phytochrome (PHY) photoreceptors and acts as a key repressor of cell differentiation that negatively regulates stomatal development (Figure 6a).”
